# BATCHED BAYESIAN OPTIMIZATION IN DISCRETE DOMAINS BY MAXIMIZING THE PROBABILITY OF INCLUDING THE OPTIMUM

## ABSTRACT

Batched Bayesian optimization (BO) can accelerate molecular design by efficiently identifying top-performing compounds from a large chemical library. Existing acquisition strategies for batch design in BO aim to balance exploration and exploitation. This often involves optimizing non-additive batch acquisition functions, necessitating approximation via myopic construction and/or diversity heuristics. In this work, we propose an acquisition strategy for discrete optimization that is motivated by pure exploitation, qPO (multipoint Probability of Optimality). qPO maximizes the probability that the batch includes the true optimum, which is expressible as the sum over individual acquisition scores and thereby circumvents the combinatorial challenge of optimizing a batch acquisition function. We differentiate the proposed strategy from parallel Thompson sampling and discuss how it implicitly captures diversity. Finally, we apply our method to the model-guided exploration of large chemical libraries and provide empirical evidence that it performs better than or on par with state-of-the-art methods in batched Bayesian optimization.

## 1 INTRODUCTION

Predictive modeling can greatly accelerate molecular and materials discovery. Data-driven and simulation-based models can aid in prioritizing experiments for the design of drugs (Liu et al., 2024; 2023; Horne et al., 2024) and other materials (Schwalbe-Koda et al., 2021; Gómez-Bombarelli et al., 2016). In *iterative* design cycles guided by predictive models, the reliability of model predictions can be gradually improved as the model is updated with newly collected data. This iterative, model-guided approach has been applied to the discovery of drugs (Desai et al., 2013), laser emitters (Strieth-Kalthoff et al., 2024), and dyes (Koscher et al., 2023; Bassman Oftelie et al., 2018).

Bayesian optimization (BO) is arguably the most popular mathematical framework for iterative model-based design (Frazier, 2018; Garnett, 2023). BO optimizes an expensive black-box objective function ($f$ or "oracle") by iteratively training a surrogate model and using its predictions to select designs for evaluation (Frazier, 2018). At each iteration, an *acquisition function* uses the mean and/or uncertainty of surrogate model predictions to select which design(s) to evaluate. BO has been applied to efficiently explore chemical libraries in numerous previous works (Cherkasov et al., 2006; Graff et al., 2021; Yang et al., 2021; Bellamy et al., 2022; Wang-Henderson et al., 2023).

Computational (e.g., physics-based simulations) and experimental (e.g., bioactivity assays) oracles in many chemistry applications are most efficiently evaluated in parallel. The total evaluation budget is therefore spread across relatively few iterations with large batch sizes, requiring an acquisition function to select a *batch* of experiments. The non-additivity of batch-level acquisition functions complicates the selection of optimal batches in BO; when the value of selecting a candidate depends on other selections, batch design becomes a combinatorial problem. Optimizing Bayes-optimal batch acquisition functions is therefore often computationally intractable even for modest batch sizes (Garnett, 2023, Section 11.3). Strategies that fail to consider this non-additivity, such as selecting the top candidates based on a sequential policy, can produce homogeneous batches that lack diversity. Prior works have primarily relied on (1) methods to increase diversity (Gonzalez et al., 2016; Kathuria et al., 2016; Nguyen et al., 2016; Groves & Pyzer-Knapp, 2018), (2) hallucinated

observations to approximate intractable integrals (Ginsbourger et al., 2010; Desautels et al., 2014), (3) the randomness inherent to Thompson sampling (Thompson, 1933) to extend it to the batch setting (Hernández-Lobato et al., 2017; Dai et al., 2022), or (4) some combination thereof (Ren & Li, 2024; Nava et al., 2022).

In this paper, we propose qPO (multipoint Probability of Optimality), a batch construction strategy that maximizes the likelihood that the optimum exists in the acquired batch. Inspired by parallel Thompson sampling (Hernández-Lobato et al., 2017; Kandasamy et al., 2018), qPO centers around the probability of optimality, accounts for correlations between inputs, and is naturally parallelizable. However, qPO aims to forego randomness. While this distinction may seem to hinder exploration, the consideration of a surrogate model's joint distribution over all candidates allows qPO to favor diversity. Uniquely, the defined batch-level acquisition function can be expressed as a sum of individual candidate acquisition scores, circumventing the combinatorial challenge of maximizing a batch-level acquisition function. We summarize the contributions of this work as follows:

1. We present a novel exploitative strategy for batch design in discrete Bayesian optimization that maximizes the likelihood of including the true optimum in the batch.

2. We derive a batch-level acquisition function that is equal to the sum of individual acquisition scores and is thereby maximized by selecting the top candidates by acquisition score.

3. Through a simple analytical case study, we demonstrate the importance of considering prediction covariance in exploitation, describe how covariance can capture diversity, and differentiate our method from parallel Thompson sampling.

4. We demonstrate that our acquisition strategy identifies top-performers from chemical libraries as efficiently as state-of-the-art alternatives for batched BO in two realistic molecular discovery settings.

## 2 MAXIMIZING THE PROBABILITY OF INCLUDING THE OPTIMUM

### 2.1 PRELIMINARIES

We first assume that there exists an expensive black box oracle function $f(\cdot)$ that maps each candidate $x_i$ to a scalar objective value $y_i$. Following Hernández-Lobato et al. (2017), we assume that evaluations of $f$ are noise-free. Our aim is to solve the following optimization problem:

$$x^* = \underset{x \in \mathcal{X}}{\arg\max} \ f(x), \tag{1}$$

where $\mathcal{X}$ is a fixed discrete design space, e.g., of molecular structures from a virtual library comprised of $N$ candidates $\{x_i\}$ for $i = 1, ..., N$.

Bayesian optimization (BO) aims to solve this optimization using an iterative model-guided approach. In each iteration, we select $b$ candidates for parallel evaluation with $f$. We denote the set of acquired candidates $\mathcal{X}_{acq}$. In each iteration, $f$ is evaluated for all $x_i$ in $\mathcal{X}_{acq}$, and a surrogate model $\hat{f}$ that predicts $f$ is (re)trained with newly acquired data. An acquisition function utilizes surrogate model predictions on candidates $x_i$ for $i = 1, ..., N$ to select the next set of evaluations. The iterative procedure ends when some stopping criterion is met: computational or experimental resources are expended, a maximum number of iterations is reached, or a satisfactory value of $f$ is achieved.

The imperfect surrogate model $\hat{f}$ provides a probabilistic prediction $y$ of the objective function value(s) for one or more candidates. This distribution over possible values of $y$ for candidate $x$, denoted $p(y|x)$, may be described by a machine learning model such as a Gaussian process or Bayesian neural network. We may alternately have a surrogate model that does not form a continuous probability distribution but still enables sampling, e.g., through deep ensembling (Lakshminarayanan et al., 2017) or Monte Carlo dropout (Gal & Ghahramani, 2016). Without loss of generality, we consider the prediction to be an integral over possible model parameters $\boldsymbol{\theta}$:

$$p(y|x) = \int_{\boldsymbol{\theta}} p(y|x, \boldsymbol{\theta}) p(\boldsymbol{\theta}) d\boldsymbol{\theta}, \tag{2}$$

where $p(\boldsymbol{\theta})$ does not truly have to represent a prior, but can represent a posterior distribution given some training data. We consider all discrete candidates $x_i \in \mathcal{X}$ to be deterministic and fixed.

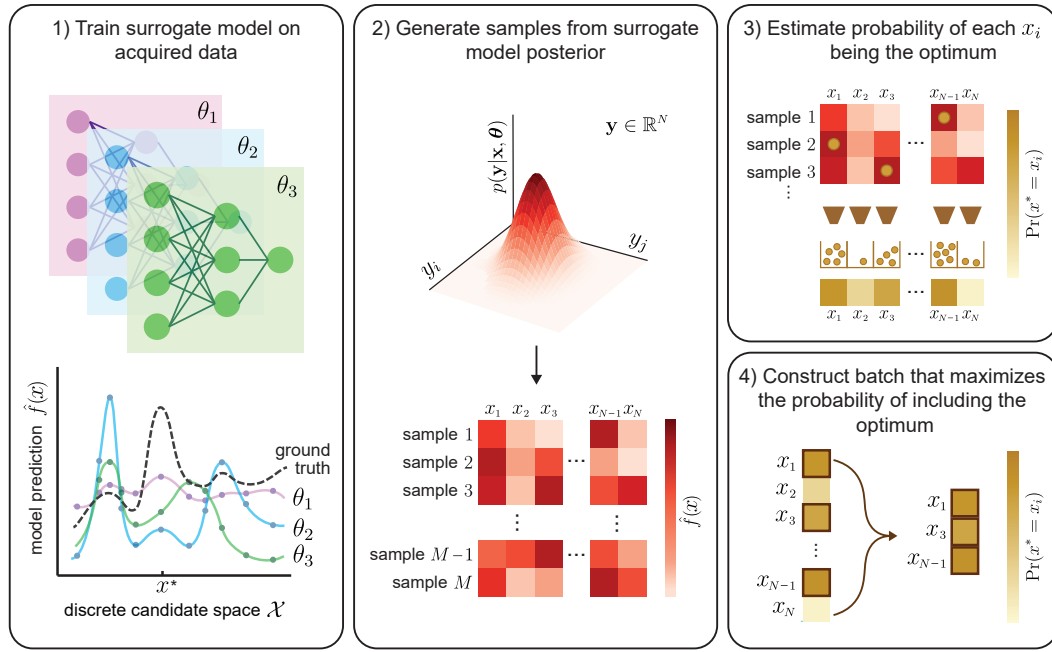

Figure 1: Exploitative batch design by maximizing the likelihood of including the optimum, in the context of a single Bayesian optimization iteration. First, a probabilistic surrogate model is trained on acquired data. Second, samples are obtained from the joint posterior distribution over all candidates. When direct posterior sampling is impossible or inefficient, a multivariate Gaussian may be modeled from the true posterior to enable approximate posterior sampling. Third, we estimate from these samples the probability that each candidate is the true optimum. Fourth, the batch is populated with candidates most likely to be optimal; in doing so, the proposed strategy maximizes the probability that the batch contains the true optimum. In addition to the sampling-based approach visualized here, we describe alternative methods to approximate acquisition scores in Section 2.3.2.

## 2.2 DERIVING AN ACQUISITION FUNCTION FOR OPTIMAL BATCH DESIGN

Most acquisition strategies in BO aim to balance exploitation and exploration. Exploitation prioritizes selections that are most likely to achieve the highest oracle score, while exploration is intended to prevent a search from getting "stuck" in local optima. Conceptually, exploration is expected to contribute to a more reliable surrogate model and thereby benefit the optimization in the long run; a classic failure mode of BO within continuous design spaces is the oversampling of a single mode of the surrogate model posterior (Hernández-Lobato et al., 2014). In contrast, exploitation selects the best candidates at a given iteration without consideration of the impact on future iterations. We pursue a batch acquisition strategy that is motivated by exploitation, optimizing expected performance in the immediate iteration as if the optimization could be stopped at any time.[1] Our acquisition strategy is visualized within the context of one BO iteration in Figure 1.

We aim to maximize the likelihood that the true optimum $x^*$ exists in the acquired batch $\mathcal{X}_{acq}$. An optimal batch of size $b$ solves the optimization:

$$\mathcal{X}_{acq}^* = \underset{\mathcal{X}_{acq} \subset \mathcal{X}, |\mathcal{X}_{acq}| = b}{\arg\max} \Pr(x^* \in \mathcal{X}_{acq}). \tag{3}$$

We focus on continuous surrogate models with low but non-zero observation noise (e.g., Gaussian processes), such that the probability of two inputs having the exact same output is 0. Therefore, we may assume that the events $\{\hat{f}(x^*) = \hat{f}(x_i)\}_{x_i \in \mathcal{X}}$ are mutually exclusive. In our estimation of Eq. 3, we will treat $\{\hat{f}(x^*) = \hat{f}(x_i)\}$ as equivalent to $\{x^* = x_i\}$, leading to the presumed mutual

---

[1]This is a realistic setting for molecular discovery campaigns. When the cost of evaluating $f$ varies across compounds in the design space, the number of iterations or oracle budget may be uncertain when the optimization begins.

exclusivity of the events $\{x^* = x_i\}_{x_i \in \mathcal{X}}$ and the following result:

$$\Pr(x^* \in \mathcal{X}_{acq}) = \sum_{x_i \in \mathcal{X}_{acq}} \Pr(x^* = x_i) \tag{4}$$

$$= \sum_{x_i \in \mathcal{X}_{acq}} P(f(x^*) = f(x_i)) \tag{5}$$

$$= \sum_{x_i \in \mathcal{X}_{acq}} P(f(x_i) > f(x_j) \,\forall\, j \neq i). \tag{6}$$

We elaborate on scenarios where a unique optimum cannot be assumed in Appendix A.1. An appropriate strategy to optimize the objective in Eq. 4 is to approximate $\Pr(x^* = x_i)$ for each candidate $x_i$ and select the top $b$ candidates based on the resulting approximations. We refer to this acquisition strategy as qPO (multipoint Probability of Optimality), which is deceptively simple to apply in practice. Consider a candidate $x_i$ in a list of candidates $\mathbf{x}$—designs that have not been previously acquired. Using the corresponding predictions $\mathbf{y}$ modeled by the surrogate posterior, we may estimate each $\Pr(x^* = x_i)$ as the following integral or expectation:

$$\alpha_i = \int_{\mathbb{R}^N} \mathbb{1}_{y_i = \max(\mathbf{y})} \, p(\mathbf{y}|\mathbf{x}) \, d\mathbf{y} \tag{7}$$

$$= \mathbb{E}_{\mathbf{y} \sim p(\mathbf{y}|\mathbf{x})} \left[ \mathbb{1}_{y_i = \max(\mathbf{y})} \right], \tag{8}$$

where each $\alpha_i$ represents an acquisition score for $x_i$. A special outcome of this batch design strategy is that qPO is naturally a sum over individual acquisition scores. Batch acquisition functions that are extensions of sequential ones, such as multipoint expected improvement (qEI) (Ginsbourger et al., 2010), typically require evaluation of the entire batch at once. Myopic construction of large batches can only approximately optimize these batch-level acquisition functions, risking the selection of suboptimal batches. In contrast, qPO can be optimized without the issue of non-additivity and resulting approximations faced by many myopic batch construction strategies.

### 2.3 METHODS TO APPROXIMATE ACQUISITION SCORES NUMERICALLY

#### 2.3.1 MONTE CARLO INTEGRATION TO APPROXIMATE ACQUISITION SCORES

The expectation in Eq. 8 may be approximated with $M$ samples from the surrogate model posterior:

$$\alpha_i \approx \frac{1}{M} \sum_{m=1}^{M} \mathbb{1}_{y_i^{(m)} = \max(\mathbf{y}^{(m)})}, \tag{9}$$

where $\mathbf{y}^{(m)} = [y_1^{(m)}, y_2^{(m)}, ..., y_N^{(m)}]$ is the $m$th sample from the surrogate model posterior for $N$ candidates. In practice, these may be samples from the posterior of a Gaussian process, predictions based on sampled parameters $\boldsymbol{\theta}^{(m)}$ from a Bayesian neural network, or predictions based on Monte Carlo dropout. The result in A.2 shows that candidates with a high probability of optimality are very likely to appear in the batch even with modest $M$, making a Monte Carlo estimate reasonable. Algorithm 1 summarizes the implementation of qPO using Monte Carlo sampling.

**Enabling efficient sampling by approximating the posterior as a multivariate Gaussian** The only requirement for this approximation of acquisition scores is that the posterior can be sampled. However, efficient and widespread algorithms exist to sample from Gaussian distributions (Vono et al., 2022; Aune et al., 2013). The computational cost of posterior sampling may be reduced by sampling from a multivariate Gaussian that approximates the true posterior. This approximate posterior may be obtained from an arbitrary number of approximate samples from the true posterior. For example, one might obtain a multivariate Gaussian posterior based on the predictions of a neural network ensemble; this could apply for an ensemble of any model architecture and for an arbitrary number of models in the ensemble. Specifically, given a smaller number ($< M$) of samples, one can construct an empirical mean and covariance matrix from which additional samples can be drawn.

---

**Algorithm 1** Bayesian optimization with qPO using Monte Carlo integration

---

**Input:** design space $\mathcal{X}$, oracle function $f$, initial data $\mathcal{D}_0$, batch size $b$, number of samples $M$, number of iterations $T$
**for** $t = 1$ **to** $T$ **do**
    Compute joint posterior $p(\mathbf{y}|\mathbf{x}, \boldsymbol{\theta})$ over unacquired candidates $\mathbf{x} \in \mathcal{X}$
    **for** $m = 1$ **to** $M$ **do**
        $\mathbf{y}^{(m)} \sim p(\mathbf{y}|\mathbf{x}, \boldsymbol{\theta})$                                      $\triangleright$ Sample from joint posterior
    **end for**
    **for** $i = 1$ **to** $|\mathbf{x}|$ **do**
        $\alpha_i \leftarrow \frac{1}{M} \sum_{m=1}^{M} \mathbb{1}_{y_i^{(m)} = \max(\mathbf{y}^{(m)})}$                $\triangleright$ Compute qPO acquisition scores
    **end for**
    $\mathcal{X}_{acq}^* \leftarrow \text{top-}k(\{x_i, \alpha_i\}_{i=1,\ldots,|\mathbf{x}|}, b)$         $\triangleright$ Select $b$ candidates with greatest qPO scores
    Evaluate $f(x_i)$ for all $x_i \in \mathcal{X}_{acq}^*$                         $\triangleright$ Call the oracle
    $\mathcal{D}_t \leftarrow \mathcal{D}_{t-1} \cup \{x_i, y_i\}_{x_i \in \mathcal{X}_{acq}^*}$                     $\triangleright$ Update training data
**end for**
**Return:** Acquired data $\mathcal{D}_T$

---

**Coping with low probability events** If very few candidates are perceived by the model as having a substantial probability of being optimal, $\Pr(x^* = x_i)$ may be estimated to be zero for many candidates for finite $M$. Filling a batch of size $b$ may emerge as a challenge if there are fewer than $b$ non-zero acquisition scores. A simple way to address this issue is to fill the remainder of the batch using an alternative metric (e.g., greedy or upper confidence bound). Methods designed for rare event estimation (de Boer et al., 2005; Cérou et al., 2012; Gibson & Kroese, 2022) may alternatively be implemented to assign acquisition scores to candidates with small probabilities of optimality.

### 2.3.2 ADDITIONAL STRATEGIES FOR NUMERICAL APPROXIMATION OF QPO ACQUISITION SCORES

Techniques beyond Monte Carlo integration may also enable the estimation of qPO. If the posterior is a multivariate Gaussian, the acquisition score in Eq. 6 can be recast as an orthant probability through a change of variables, providing an alternative approach to predicting $\Pr(x^* = x_i)$ that does not rely on posterior sampling (Azimi et al., 2010; Azimi, 2012). This orthant probability, defined in A.3, may be approximated analytically using a whitening transformation (Azimi, 2012). Additionally, Cunningham et al. (2013) propose an expectation propagation approach (Minka, 2001) to directly approximate orthant probabilities, and Gessner et al. (2020) introduce an integrator for truncated Gaussians that accurately estimates even small Gaussian probabilities. The orthant probabilities may also be estimated using numerical integration, which involves Cholesky decomposition followed by Monte-Carlo sampling (Genz, 1992). Alternative methods to estimate high-dimensional orthant probabilities (Miwa et al., 2003; Craig, 2008; Ridgway, 2016) may also be applied to the estimation of qPO acquisition scores. We consider the implementation of these methods future work.

## 3 RELATED WORK

### 3.1 BATCHED BAYESIAN OPTIMIZATION

A naive batch construction strategy is to select the top $b$ compounds based on an acquisition function designed for sequential BO. However, because the utility of a given selection depends on other selections, the top-$b$ approach does not guarantee an optimal batch in general (Garnett, 2023).

One general approach to batch design is to, as closely as possible, replicate the behavior of a sequential policy. Azimi et al. (2010) define an acquisition function that minimizes the discrepancy between the batch policy and sequential policy behavior. Batches may alternatively be constructed iteratively (myopically) by hallucinating the outcomes of previous selections in the batch. This enables the optimization of batch-level acquisition functions like multipoint expected improvement (Ginsbourger et al., 2010) and batch upper confidence bound (Desautels et al., 2014). However, this approach is typically limited to cases where the surrogate model can efficiently be updated with hallucinated

or pending data. Further, batches that are constructed myopically only approximately optimize the batch-level acquisition function. In contrast, our approach does not rely on hallucination and can be applied to any posterior which can be sampled or modeled as a multivariate Gaussian.

Methods to improve diversity have also been applied to prevent the selection of candidates that would provide minimal marginal information gain. Gonzalez et al. (2016) use local penalization to construct diverse batches. Determinantal point processes (DPPs) have also been used for batch diversification in discrete optimization (Kathuria et al., 2016); Nava et al. (2022) demonstrate improved theoretical convergence rates by incorporating DPPs into parallel Thompson sampling. Nguyen et al. (2016) and Groves & Pyzer-Knapp (2018) model the objective landscape as mixture of Gaussians and acquire predicted local optima. Strategies based on diversity heuristics generally assume that diverse batches support exploration, not necessarily exploitation. We describe in Section 3.2 how our method does capture diversity, even with exploitation as the primary motivation.

Sequential acquisition functions that randomly select candidates by sampling, like Thompson sampling, can be extended to the batch case by increasing the number of random samples. Parallel Thompson sampling involves sampling from the model posterior and acquiring the optimum point from each sample (Hernández-Lobato et al., 2017). TS-RSR uses a similar methodology, but selects from each sample the point that minimizes a regret to uncertainty ratio (Ren & Li, 2024). Dai et al. (2022) extend neural Thompson sampling (Zhang et al., 2020) to the batch setting; each of $b$ randomly initialized neural networks determines one selection in the batch. The randomness inherent to Thompson sampling and related strategies is expected to contribute to the search's exploration. qPO, in contrast, is intended to make selections deterministically.

### 3.2 COMPARISON WITH ALTERNATIVE BATCH STRATEGIES

We continue with an illustrative example to highlight how the proposed approach captures diversity, a key contribution to qPO's robustness. Consider the following predictive distribution for $\mathbf{y}$ given $\mathbf{x}$:

$$\mathbf{y} \sim \mathcal{N}\left( \begin{bmatrix} 10 \\ 5 \\ 0 \end{bmatrix}, \begin{bmatrix} 101 & 100 & 0 \\ 100 & 101 & 0 \\ 0 & 0 & 1 \end{bmatrix} \right) \tag{10}$$

The probabilities of $x_1$, $x_2$, and $x_3$ being optimal are roughly 84%, 0%, and 16%, respectively. For $b = 2$, our acquisition strategy would select $x_1$ and $x_3$, while a greedy strategy based only on the posterior mean vector would select $x_1$ and $x_2$. A diversity-aware acquisition strategy would select $x_1$ and $x_3$ if we assume that the high covariance between $x_1$ and $x_2$ reflects design space similarity. This assumption is particularly valid for molecular applications when the surrogate model is a Gaussian process with a Tanimoto kernel. Because the Tanimoto kernel defines prior covariance based on structural similarity, structurally similar compounds will have higher covariance. Our method naturally captures this sense of diversity through model covariance without requiring clustering or other definitions of diversity that are not inherent to the model itself.

When Monte Carlo integration is used to approximate qPO, our method resembles Thompson sampling, particularly the parallel Thompson sampling (pTS) approach proposed by Hernández-Lobato et al. (2017) and Kandasamy et al. (2018). Both methods choose a batch of $b$ inputs by selecting the maxima of posterior samples. Our approach differs from parallel Thompson sampling in two notable ways. First, in the case of $b = 1$, Thompson sampling chooses candidates *randomly* with probability *proportional* to $\Pr(x^* = x_i)$. qPO aims to choose candidates *deterministically* that *maximize* $\Pr(x^* = x_i)$. However, because a closed-form solution for qPO acquisition scores does not exist, the implementation of our acquisition strategy is not purely deterministic. Second, qPO generalizes to the $b \geq 2$ case differently from pTS. As described in Algorithm 2 of Hernández-Lobato et al. (2017), if the optimum input for a particular sample is already in the batch, then the second most optimal input is added (or the third if the second most optimal input is also in the batch, and so forth). If the model makes highly correlated predictions, this will result in a batch filled with candidates that have highly correlated predictions. In the analytical example shown above in Eq. 10, pTS will be more likely to select $x_1$ and $x_2$ than $x_1$ and $x_3$. In contrast, once our method has added a first candidate to the batch, subsequent selections are chosen conditioned on existing selections not being optimal.

## 4 EXPERIMENTS

### 4.1 BASELINES AND EVALUATION METRICS

We apply qPO to two model-guided searches of chemical libraries and compare its performance to alternative batch acquisition functions: pTS (Hernández-Lobato et al., 2017; Kandasamy et al., 2018), Determinantal Point Process Thomspon Sampling (DPP-TS) (Nava et al., 2022), Thompson Sampling with Regret to Sigma Ratio (TS-RSR) (Ren & Li, 2024), General-purpose Information-Based Bayesian OptimisatioN (GIBBON) (Moss et al., 2021), multipoint probability of improvement (qPI), multipoint expected improvement (qEI), batch upper confidence bound (BUCB) (Wilson et al., 2017), upper confidence bound (UCB), and greedy (mean only). DPP-TS and TS-RSR are extensions of pTS with improved theoretical guarantees. BUCB, qPI, and qEI are batch-level extensions of sequential policies that employ myopic construction in finite discrete domains. GIBBON is an information-based batch acquisition function. We analyze the retrieval of the true top-$k$ acquired, the average oracle value of the acquired top-$k$, and cumulative regret, the former two being common metrics for assessing model-guided virtual screening methods (Pyzer-Knapp, 2018; Graff et al., 2021; Wang-Henderson et al., 2023). Run times are reported in A.5.

Both demonstrations use a Tanimoto Gaussian process surrogate model with a constant mean that operates on 2048-length count Morgan fingerprints. qPO is implemented following Eq. 9 using $M = 10,000$. Pairs of candidates $\{x_i, x_j\}$ for which $\alpha_i = \alpha_j$ are ranked by their predicted mean (A.4.1). The cost of sampling from a multivariate Gaussian for $N$ candidates is $O(N^3)$ due to the Cholesky decomposition (Vono et al., 2022). To alleviate the computational cost of sampling-based methods—pTS, DPP-TS, TS-RSR, GIBBON, qPI, qEI, BUCB, and qPO—for large values of $N$, in each iteration we reduce the set of $N$ candidates to 10,000 using a greedy metric. We then apply the respective strategy to select a batch from the smaller set of candidates. For qPO, this modifies the set of candidates $\mathbf{x}$ considered in Algorithm 1. Because this may impact the exploration characteristics of pTS and qEI, we include an additional baseline ("random10k") that randomly selects compounds from these top 10,000 candidates. A.6 compares qPO performance when using this pre-filtering strategy with pre-filtering based on UCB. Experimental details can be found in A.4.

### 4.2 APPLICATION TO ANTIBIOTIC DISCOVERY

Our first demonstration applies Bayesian optimization to the retrospective identification of putative antibiotics with activity against *Staphylococcus aureus*. Wong et al. (2024) experimentally screened 39,312 compounds for growth inhibition of *S. aureus*. We search this dataset for compounds with the lowest reported mean growth of *S. aureus*, indicating greatest antibiotic activity. For each run, we randomly select an initial batch of 50 compounds and select 50 more in each of 10 iterations.

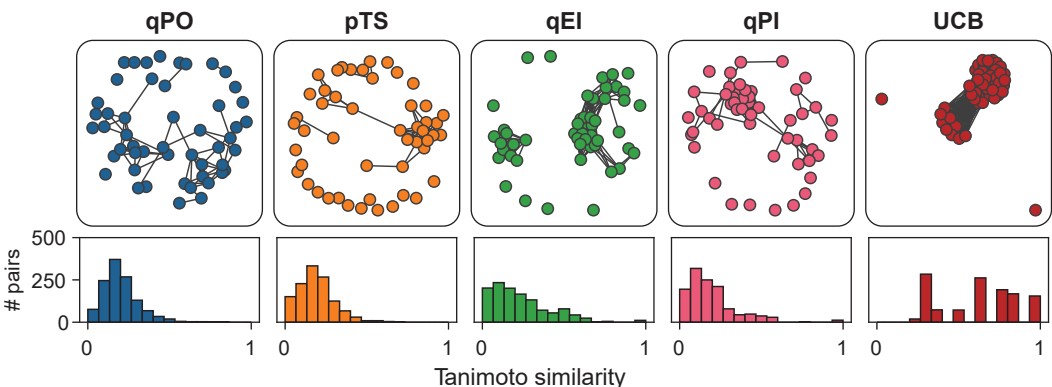

Figure 2: Batch diversity of a model-guided optimization loop for antibiotic discovery. Networks depict selected batches in the first iteration after training on a randomly selected (seed of 7) initial batch of 50 designs with growth inhibition values from Wong et al. (2024). Nodes represent acquired compounds; edges are drawn between pairs with Tanimoto similarity > 0.4. Nodes are positioned using the Fruchterman-Reingold force-directed algorithm (Fruchterman & Reingold, 1991). Histograms portray the distribution of Tanimoto similarity scores for all pairs in the selected batch.

| Method | Iteration | Top 10 Average (↓) | Top 100 Average (↓) | Fraction Top 0.5% (↑) | Fraction Top 1% (↑) | Cumulative Regret (↓) |
|---|---|---|---|---|---|---|
| BUCB | 5 | 0.15 ± 0.02 | 0.51 ± 0.07 | 0.05 ± 0.01 | 0.06 ± 0.01 | 0.58 ± 0.14 |
| DPP-TS | 5 | 0.15 ± 0.00 | 0.71 ± 0.01 | 0.02 ± 0.00 | 0.02 ± 0.00 | 0.53 ± 0.14 |
| GIBBON | 5 | 0.18 ± 0.02 | 0.69 ± 0.04 | 0.02 ± 0.01 | 0.03 ± 0.01 | 0.57 ± 0.16 |
| Greedy | 5 | 0.18 ± 0.06 | 0.45 ± 0.07 | 0.05 ± 0.01 | 0.06 ± 0.01 | 1.01 ± 0.35 |
| TS-RSR | 5 | 0.18 ± 0.05 | 0.47 ± 0.08 | 0.05 ± 0.01 | 0.06 ± 0.01 | 0.75 ± 0.22 |
| UCB | 5 | 0.19 ± 0.05 | 0.49 ± 0.08 | 0.05 ± 0.01 | 0.06 ± 0.01 | 1.01 ± 0.34 |
| pTS | 5 | 0.17 ± 0.04 | 0.62 ± 0.06 | 0.04 ± 0.01 | 0.05 ± 0.01 | 0.55 ± 0.12 |
| qEI | 5 | 0.19 ± 0.04 | 0.62 ± 0.07 | 0.03 ± 0.01 | 0.04 ± 0.01 | 0.74 ± 0.25 |
| qPI | 5 | 0.14 ± 0.03 | 0.43 ± 0.07 | 0.06 ± 0.01 | 0.08 ± 0.01 | 0.72 ± 0.24 |
| qPO | 5 | 0.18 ± 0.06 | 0.46 ± 0.06 | 0.06 ± 0.01 | 0.07 ± 0.01 | 0.69 ± 0.29 |
| random10k | 5 | 0.22 ± 0.02 | 0.80 ± 0.01 | 0.01 ± 0.00 | 0.01 ± 0.00 | 0.63 ± 0.15 |
| BUCB | 10 | 0.12 ± 0.02 | 0.27 ± 0.07 | 0.10 ± 0.02 | 0.13 ± 0.02 | 0.64 ± 0.15 |
| DPP-TS | 10 | 0.13 ± 0.00 | 0.55 ± 0.02 | 0.04 ± 0.00 | 0.04 ± 0.00 | 0.59 ± 0.14 |
| GIBBON | 10 | 0.13 ± 0.01 | 0.51 ± 0.06 | 0.05 ± 0.01 | 0.05 ± 0.01 | 0.64 ± 0.17 |
| Greedy | 10 | 0.11 ± 0.00 | 0.21 ± 0.02 | 0.11 ± 0.01 | 0.14 ± 0.02 | 1.07 ± 0.35 |
| TS-RSR | 10 | 0.11 ± 0.00 | 0.20 ± 0.03 | 0.12 ± 0.01 | 0.16 ± 0.02 | 0.81 ± 0.22 |
| UCB | 10 | 0.11 ± 0.00 | 0.24 ± 0.05 | 0.12 ± 0.02 | 0.15 ± 0.02 | 1.07 ± 0.34 |
| pTS | 10 | 0.11 ± 0.01 | 0.29 ± 0.06 | 0.11 ± 0.02 | 0.13 ± 0.02 | 0.61 ± 0.12 |
| qEI | 10 | 0.12 ± 0.01 | 0.32 ± 0.06 | 0.08 ± 0.01 | 0.11 ± 0.02 | 0.80 ± 0.25 |
| qPI | 10 | 0.10 ± 0.00 | 0.17 ± 0.01 | 0.14 ± 0.01 | 0.17 ± 0.01 | 0.76 ± 0.24 |
| qPO | 10 | 0.11 ± 0.01 | 0.21 ± 0.05 | 0.14 ± 0.02 | 0.19 ± 0.03 | 0.73 ± 0.30 |
| random10k | 10 | 0.15 ± 0.00 | 0.68 ± 0.01 | 0.02 ± 0.00 | 0.02 ± 0.00 | 0.71 ± 0.15 |

Table 1: Optimization performance for the iterative discovery of antibiotic compounds at an intermediate and final iteration. The average oracle value of the top 10 and 100 acquired designs are shown, where a lower value indicates greater antibiotic activity. We also report retrieval of the true top 0.5% and 1%, representing 197 and 393 top-performing compounds, respectively, as well as cumulative regret. All values denote averages ± one standard error of the mean across ten runs.

Optimization performance of qPO and baselines is shown in Table 1. While no single strategy outperforms all others in all metrics, qPO consistently performs on par with state-of-the-art baselines, with qPI appearing to be the most competitive baseline in this experiment. The retrieval of the top 0.5% is plotted for all iterations in Figure 3A for qPO and competitive baselines.

We also analyze batch diversity to assess whether qPO captures diversity in this empirical setting. The diversity of acquired batches across select strategies is visualized in Figure 2 (details in A.4.4). qPO, pTS, and qPI appear to obtain the most diverse selections, with UCB selecting the least diverse batch. While this visualization depicts acquired batches in a single iteration, these results indicate that qPO can achieve diversity without imposing randomness, myopic construction, or diversity heuristics.

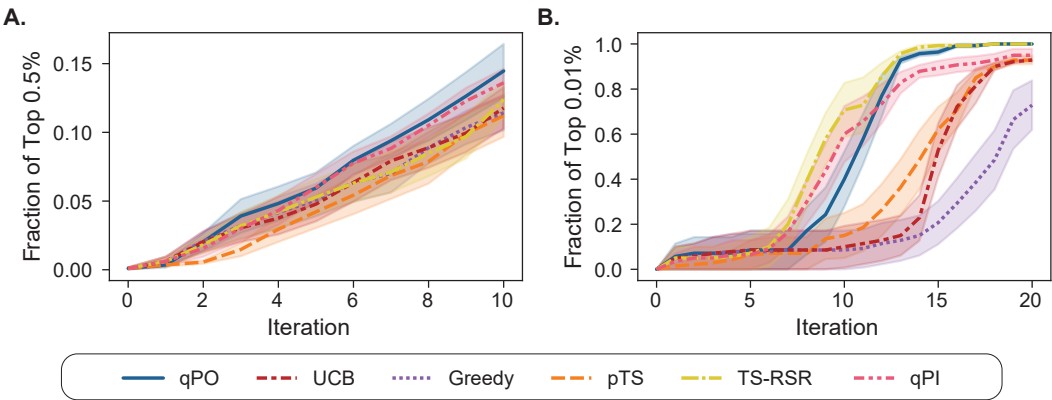

Figure 3: Retrieval profile for two model-guided searches of chemical libraries. (A) Retrieval of the top 0.5% (197) designs for the iterative discovery of putative antibiotics (Section 4.2). (B) Retrieval of the top 0.01% (14) designs for the iterative discovery of organic materials (Section 4.3). For visibility, top-performing methods based on Tables 1 and 2 were selected for visualization. qPO performs on par with state-of-the-art methods for both case studies. qPI is competitive in both case studies, while TS-RSR is competitive primarily in the second case study (B). Shaded regions denote ± one standard error of the mean across ten runs.

### 4.3 APPLICATION TO THE DESIGN OF ORGANIC ELECTRONICS

We next demonstrate qPO on the pursuit of molecules from the QM9 dataset of 133K compounds that maximize the DFT-calculated HOMO-LUMO gap (Ramakrishnan et al., 2014; Ruddigkeit et al., 2012). We begin each search with a randomized initial batch of 100 compounds and select 100 more at each of 20 subsequent iterations. Optimization performance according to top-$k$ metrics and cumulative regret is shown in Table 2, and top 0.01% retrieval of qPO and select baselines is plotted for all iterations in Figure 3B. As in the previous study, qPO performs on par with top-performing baselines across metrics. We observe the greatest improvement to top-$k$ optimization performance when considering small values of $k$ (Figure 3), aligning with qPO's primary focus of identifying the true global optimum.

| Method | Iteration | Top 10 Average ($\uparrow$) | Top 100 Average ($\uparrow$) | Fraction Top 0.01% ($\uparrow$) | Fraction Top 1% ($\uparrow$) | Cumulative Regret ($\downarrow$) |
|---|---|---|---|---|---|---|
| BUCB | 10 | $0.45 \pm 0.01$ | $0.39 \pm 0.00$ | $0.80 \pm 0.07$ | $0.31 \pm 0.01$ | $1.83 \pm 0.21$ |
| DPP-TS | 10 | $0.42 \pm 0.01$ | $0.38 \pm 0.00$ | $0.41 \pm 0.12$ | $0.30 \pm 0.00$ | $2.22 \pm 0.17$ |
| Greedy | 10 | $0.40 \pm 0.01$ | $0.38 \pm 0.00$ | $0.09 \pm 0.09$ | $0.36 \pm 0.02$ | $2.32 \pm 0.21$ |
| TS-RSR | 10 | $0.45 \pm 0.01$ | $0.39 \pm 0.00$ | $0.71 \pm 0.12$ | $0.36 \pm 0.01$ | $2.03 \pm 0.15$ |
| UCB | 10 | $0.40 \pm 0.01$ | $0.38 \pm 0.00$ | $0.10 \pm 0.09$ | $0.37 \pm 0.01$ | $2.32 \pm 0.21$ |
| pTS | 10 | $0.40 \pm 0.01$ | $0.38 \pm 0.00$ | $0.15 \pm 0.10$ | $0.23 \pm 0.01$ | $2.42 \pm 0.19$ |
| qEI | 10 | $0.43 \pm 0.01$ | $0.38 \pm 0.00$ | $0.46 \pm 0.14$ | $0.28 \pm 0.02$ | $2.07 \pm 0.20$ |
| qPI | 10 | $0.44 \pm 0.01$ | $0.39 \pm 0.00$ | $0.60 \pm 0.12$ | $0.27 \pm 0.01$ | $2.13 \pm 0.10$ |
| qPO | 10 | $0.42 \pm 0.01$ | $0.38 \pm 0.00$ | $0.40 \pm 0.12$ | $0.35 \pm 0.01$ | $2.23 \pm 0.21$ |
| random10k | 10 | $0.38 \pm 0.00$ | $0.36 \pm 0.00$ | $0.01 \pm 0.01$ | $0.07 \pm 0.00$ | $2.62 \pm 0.05$ |
| BUCB | 20 | $0.47 \pm 0.00$ | $0.40 \pm 0.00$ | $1.00 \pm 0.00$ | $0.54 \pm 0.00$ | $1.95 \pm 0.26$ |
| DPP-TS | 20 | $0.46 \pm 0.00$ | $0.40 \pm 0.00$ | $0.94 \pm 0.01$ | $0.59 \pm 0.01$ | $2.50 \pm 0.24$ |
| Greedy | 20 | $0.45 \pm 0.01$ | $0.39 \pm 0.00$ | $0.73 \pm 0.11$ | $0.65 \pm 0.01$ | $3.73 \pm 0.39$ |
| TS-RSR | 20 | $0.47 \pm 0.00$ | $0.40 \pm 0.00$ | $1.00 \pm 0.00$ | $0.63 \pm 0.01$ | $2.11 \pm 0.19$ |
| UCB | 20 | $0.46 \pm 0.00$ | $0.39 \pm 0.00$ | $0.93 \pm 0.00$ | $0.66 \pm 0.01$ | $3.22 \pm 0.32$ |
| pTS | 20 | $0.46 \pm 0.00$ | $0.39 \pm 0.00$ | $0.93 \pm 0.02$ | $0.50 \pm 0.01$ | $3.14 \pm 0.31$ |
| qEI | 20 | $0.46 \pm 0.00$ | $0.40 \pm 0.00$ | $0.99 \pm 0.01$ | $0.59 \pm 0.01$ | $2.25 \pm 0.26$ |
| qPI | 20 | $0.46 \pm 0.00$ | $0.39 \pm 0.00$ | $0.95 \pm 0.03$ | $0.43 \pm 0.01$ | $2.38 \pm 0.20$ |
| qPO | 20 | $0.47 \pm 0.00$ | $0.40 \pm 0.00$ | $1.00 \pm 0.00$ | $0.63 \pm 0.01$ | $2.36 \pm 0.24$ |
| random10k | 20 | $0.39 \pm 0.00$ | $0.37 \pm 0.00$ | $0.04 \pm 0.02$ | $0.15 \pm 0.00$ | $4.79 \pm 0.11$ |

Table 2: Optimization performance on the exploration of the 133K QM9 dataset for compounds with large HOMO-LUMO gaps at an intermediate and final iteration. We first report the average oracle value of the top 10 and 100 acquired designs. Retrieval of the true top 0.01% and 1% represents 14 and 1,339 top-performing compounds, respectively. All values denote average metrics $\pm$ one standard error of the mean across ten runs.

## 5 CONCLUSION

We have proposed a batch acquisition function (qPO) for discrete Bayesian optimization, motivated by exploitation, that maximizes the likelihood that the batch contains the true optimum. qPO is equal to the sum over individual acquisition scores and therefore circumvents the combinatorial challenge of optimizing a batch-level acquisition score. We explain how the treatment of model covariance implicitly captures diversity and how it differentiates qPO from parallel Thompson sampling in subtle but meaningful ways. Empirically, our method efficiently identifies an equivalent or higher percentage of top-performing candidates when compared to batched BO alternatives. The most notable improvement to top-$k$ metrics is observed for smaller values of $k$, consistent with the acquisition strategy's goal of acquiring the true global optimum.

The proposed batch acquisition strategy has some notable limitations. First, qPO cannot be computed analytically (in an exact sense), necessitating a Monte Carlo estimate. Alternative methods for estimating the probability of a candidate's optimality, such as expectation propagation, may reduce the computational cost of implementing qPO. Second, qPO may fail to select diverse batches in the presence of high observation noise, which will reduce the relative impact of covariance on qPO scores[2]. Theoretical analysis of the exploration-exploitation trade-off in the finite iteration setting may uncover potential failure modes of qPO and further support the empirical results observed here.

---

[2]For example, if $\lambda I$ were added to the covariance matrix in Eq. 10, $(x_1, x_2)$ would be the optimal batch for sufficiently large $\lambda$.

## REPRODUCIBILITY STATEMENT

All code required to generate the results shown in this work, including installation instructions, the datasets explored in Section 4, and code to run BO with qPO, is attached as supplementary material.

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

# A APPENDIX

## A.1 THE NON-UNIQUE OPTIMUM CASE

In some cases, the mutual exclusivity of events $\{x^* = x_i\}_{x_i \in \mathcal{X}}$ may not be assumed. In such scenarios, we will not arrive at the result in Eq. 4 and must instead optimize the acquisition function by constructing batches myopically (i.e., one-by-one):

$$
\begin{aligned}
\hat{x}^{*,1} &= \underset{x \in \mathcal{X}}{\arg\max} \ \Pr(x^* = x) \\
\hat{x}^{*,2} &= \underset{x \in \mathcal{X} \setminus \{\hat{x}^{*,1}\}}{\arg\max} \ \Pr(x^* = x | x^* \neq \hat{x}^{*,1}) \\
\hat{x}^{*,3} &= \underset{x \in \mathcal{X} \setminus \{\hat{x}^{*,1}, \hat{x}^{*,2}\}}{\arg\max} \ \Pr(x^* = x | x^* \neq \hat{x}^{*,1}, x^* \neq \hat{x}^{*,2}) \\
&\ \ \vdots \\
\hat{x}^{*,q} &= \underset{x \in \mathcal{X} \setminus \{\hat{x}^{*,1}, \dots, \hat{x}^{*,q-1}\}}{\arg\max} \ \Pr(x^* = x | x^* \neq \hat{x}^{*,1}, x^* \neq \hat{x}^{*,2}, \dots, x^* \neq \hat{x}^{*,q-1})
\end{aligned}
\tag{11}
$$

The methods described in Section 2.3 are also applicable to optimizing this batch acquisition function.

Note that, even when the events $\{x^* = x_i\}_{x_i \in \mathcal{X}}$ are not guaranteed to be mutually exclusive, observation noise in the surrogate model may allow us to assume that the events $\{\hat{f}(x^*) = \hat{f}(x_i)\}_{x_i \in \mathcal{X}}$ are mutually exclusive. In this case, we *can* apply the acquisition strategy in Eq. 4 because all *predictions* of the events $\{x^* = x_i\}_{x_i \in \mathcal{X}}$ will be mutually exclusive, and estimations of the conditional probabilities in Eq. 11 will in practice be equivalent to the corresponding unconditional probabilities.

## A.2 PROOF OF BOUND FOR EQ. 9

If
$$\Pr(x^* = x_i) \geq 1 - \delta^{\frac{1}{M}} \, ,$$
then the probability of never observing $(x^* = x_i)$ in $M$ i.i.d. Monte Carlo samples is less than $\delta$ (in equation 9).

*Proof.* The event of $x_i = x^*$ is binary and therefore has a Bernoulli distribution. For a Bernoulli distribution with expectation $q$, the probability of observing only negative events for $M$ i.i.d. samples is $(1 - q)^M$ and is decreasing with $q$. Re-arranging this gives the bound above. $\square$

This result shows that, with modest $M$, inputs $x$ which have a reasonable probability of being the optimum are very unlikely to never be observed as the optimum in some sample. For example, any candidate with at least a 0.7% chance of being the optimum has less than a 0.1% chance of never appearing as the optimum with $M = 10^3$ Monte Carlo samples.

We can use Hoeffding's inequality to get a more precise confidence interval:

**Lemma 1.** *Let $p = \Pr(x_i = x^*)$ and let $\hat{p}$ be a Monte Carlo estimate for $p$ with $M$ independent samples. Let $\epsilon = \sqrt{\frac{\log 2/\alpha}{2M}}$. Then the set $C = (\hat{p} - \epsilon, \hat{p} + \epsilon)$ is a $1 - \alpha$ confidence interval for every $p$ satisfying*
$$\Pr(p \in C) \geq 1 - \alpha \, .$$

*Proof.* From Hoeffding's inequality we have
$$\Pr(|p - \hat{p}|) \leq 2\exp(-2M\epsilon^2) \, .$$

Solving $\alpha = 2e^{-2n\epsilon^2}$ gives the desired result. $\square$

Setting $\alpha = 0.1\%$ to give a 99.9% confidence interval, if a particular candidate $x_i$ is never observed to be the optimum in $M = 10^3$ samples ($\hat{p} = 0$), then $p \in (-0.062, 0.062)$ with 99.9% probability.

## A.3 RECASTING ACQUISITION SCORES AS ORTHANT PROBABILITIES

The acquisition score in Eq. 6 can be defined as an orthant probability through a change of variables. This enables the estimation of $\Pr(x^* = x_i)$ without posterior sampling. However, efficient calculation of orthant probabilities is not possible for arbitrary probability distributions. Here, we choose to require the posterior to be a multivariate Gaussian (or approximated as such). The likelihood of candidate $x_i$ being optimal is equivalent to the probability that $y_i$ is greater than all $y_{j,j\neq i}$:

$$\Pr(x^* = x_i) = \Pr(y_i > y_1, y_i > y_2, ..., y_i > y_N) \tag{12}$$

Following the approach of Azimi et al. (2010), we denote the difference between $y_i$ and $y_j$ as $z_j^i$ and the vector of differences for candidate $i$ as $\mathbf{z}^i \in \mathbb{R}^{N-1}$. Using the transformation matrix $\mathbf{A}_i \in \mathbb{R}^{(N-1)\times N}$ as defined by Azimi et al. (2010) and Azimi (2012), we may define:

$$\mathbf{z}^i \sim \mathcal{N}\left(\mathbf{A}\boldsymbol{\mu_x}, \mathbf{A}\boldsymbol{\Sigma_x}\mathbf{A}^T\right) \tag{13}$$

where $\boldsymbol{\mu_x}$ and $\boldsymbol{\Sigma_x}$ parameterize the surrogate model posterior for candidates $\mathbf{x}$. The expression in Eq. 12 is equal to the orthant probability of $\mathbf{z}^i$, i.e., the probability that all elements of $\mathbf{z}^i$ are greater than 0. Azimi et al. (2010) apply a whitening transformation to the distribution defining $\mathbf{z}^i$, decorrelating all entries and enabling approximation of the orthant probability. Alternative methods to estimate high-dimensional orthant probabilities (Genz, 1992; Miwa et al., 2003; Craig, 2008; Ridgway, 2016) may also be applied here.

## A.4 EXPERIMENTAL DETAILS

All code required to reproduce the results for this work is attached as supplementary material.

### A.4.1 IMPLEMENTATION OF QPO ACQUISITION STRATEGY

We follow Eq. 9 using $M = 10,000$. These samples from the surrogate model posterior are used to estimate $\Pr(x^* = x_i)$ for each candidate $x_i$. When the objective is minimized, we instead estimate:

$$\Pr(x^* = x_i) \approx \frac{1}{M} \sum_{m=1}^{M} \mathbb{1}_{y_i^{(m)} = \min(\mathbf{y}^{(m)})}. \tag{14}$$

Candidates are always sorted primarily based on their probabilities of optimality. Any candidates that have identical $\Pr(x^* = x_i)$ are sorted by a greedy metric. For example, consider a maximization scenario where the $\Pr(x^* = x_i) = \Pr(x^* = x_j)$, and $\hat{f}(y_i) < \hat{f}(y_j)$. $x_j$ will be ranked above $x_i$ due to its predicted mean. All compounds with $\Pr(x^* = x_i) > 0$ will be ranked above compounds with $\Pr(x^* = x_i) = 0$. However, if there are too few candidates with $\Pr(x^* = x_i) > 0$ to fill a batch, then the batch is filled using a greedy metric on remaining candidates with $\Pr(x^* = x_i) = 0$.

### A.4.2 BAYESIAN OPTIMIZATION

We apply our batch acquisition strategy to the model-guided exploration of two discrete chemical libraries (Section 4). The first case study explores a library of 39,312 compounds for putative antibiotics that minimize the growth of *Staphylococcus aureus* (Wong et al., 2024). We randomly initialize our search with 50 compounds and their growth inhibition values and acquire batches of 50 compounds for 10 subsequent iterations. In the second case, we explore the QM9 dataset for compounds that maximize the HOMO-LUMO gap (Ramakrishnan et al., 2014; Ruddigkeit et al., 2012). Here, we use an initial batch size of 100 compounds and acquire 100 more for 20 subsequent iterations. For both case studies, ten runs were performed for each acquisition method with distinct random seeds that govern the initial batch.

Our surrogate model is a Gaussian process with a Tanimoto kernel (Tanimoto, 1958) and a constant mean, a common surrogate model architecture for molecular BO (Tripp et al., 2021; García-Ortegón et al., 2022; Gao et al., 2022). Compounds are featurized as 2048-length count Morgan fingerprints using `rdkit` (Landrum, 2024). At each iteration, the Gaussian process hyperparameters—mean, covariance scale, and likelihood noise—are optimized by maximizing the marginal log likelihood of the model over all previously acquired training data.

### A.4.3 ACQUISITION FUNCTIONS

Selecting batches with qEI, pTS, and qPO can be computationally expensive for large design spaces due to the $O(N^3)$ complexity of Cholesky decomposition for $N$ candidates. To reduce the computational cost of these acquisition functions, we first filter the set of $N$ candidates to the top 10,000 based on predicted mean and subsequently apply the appropriate acquisition function to select from these 10,000 candidates. This step is not necessary for the implementation of these acquisition functions but facilitates their use under computational resource limitations. A similar pre-filtering method has been applied by Moss et al. (2021) in a molecular discovery setting to reduce computational cost.

**Greedy** Acquisition scores for each compound $x_i$ were defined as $c * y_i$, where $c = 1$ if the objective is optimized and $c = -1$ if it is minimized. $y_i$ is the surrogate model mean prediction for compound $x_i$. The top-$b$ compounds based on acquisition score were selected in each iteration.

**Upper confidence bound (UCB)** Acquisition scores for each compound $x_i$ were defined as $c * y_i + \beta * \sigma_i$, where $c = 1$ if the objective is optimized and $c = -1$ if it is minimized. We set $\beta = 1$ for all runs. $y_i$ is the surrogate model mean prediction for compound $x_i$, and $\sigma_i$ is the prediction standard deviation. The top-$b$ compounds based on acquisition score were selected in each iteration.

**Batch upper confidence bound (BUCB)** We implement BUCB using the `qUpperConfidenceBound` (Wilson et al., 2017) and `optimize_acqf_discrete` functions in `botorch` (Balandat et al., 2020). `optimize_acqf_discrete` sequentially selects points and appends their corresponding $x$ values to the Gaussian process input for subsequent Monte Carlo samples, making this a hallucinating approach. Following Wilson et al. (2017), we set $\beta = \sqrt{3}$.

**Parallel Thompson sampling (pTS)** We follow the implementation of Hernández-Lobato et al. (2017), summarized in Algorithm 2. For each of $b$ posterior samples, the candidate $x_i$ which optimizes the objective and is not already in the batch is selected.

**Multipoint expected improvement (qEI)** We implement qEI using the `qLogExpectedImprovement` (Ament et al., 2023) and `optimize_acqf_discrete` functions in `botorch` (Balandat et al., 2020). `qLogExpectedImprovement` evaluates qEI using Monte Carlo sampling. `optimize_acqf_discrete` sequentially selects points and appends their corresponding $x$ values to the Gaussian process input for subsequent Monte Carlo samples, making this a hallucinating approach.

**Multipoint probability of improvement (qPI)** We implement qEI using the `qProbabilityOfImprovement` and `optimize_acqf_discrete` functions in `botorch` (Balandat et al., 2020). `qProbabilityOfImprovement` evaluates qPI using Monte Carlo sampling. `optimize_acqf_discrete` sequentially selects points and appends their corresponding $x$ values to the Gaussian process input for subsequent Monte Carlo samples, making this a hallucinating approach.

**Determinantal point process Thompson sampling (DPP-TS)** We follow the iterative batch construction strategy in Algorithm 1 described by Nava et al. (2022). We define the kernel as the Tanimoto similarity between 2048-length Morgan fingerprints. DPP-TS is an iterative batch construction policy; we allow for 1,000 iterations of DPP-TS batch design in each BO iteration.

**Thompson sampling with regret to sigma ratio (TS-RSR)** We follow Algorithm 1 in Ren & Li (2024) for the implementation of TS-RSR. As in pTS, we obtain $b$ posterior samples. The design which optimizes the regret to sigma ratio in each sample is acquired.

**General-purpose Information-Based Bayesian OptimisatioN (GIBBON)** We implement GIBBON (Moss et al., 2021) using the `qLowerBoundMaxValueEntropy` and `optimize_acqf_discrete` functions in `botorch` (Balandat et al., 2020).

**Additional baseline to reflect maximum exploration with filtering to top 10k** As described previously, we modify qEI, pTS, and qPO by first filtering the candidates to a set of 10,000 based on mean prediction and apply the respective acquisition strategy to the filtered set. This modification imposes a slight exploitative bias and thus may impact the exploratory nature of qEI and pTS. Therefore, we include an additional baseline that reflects the maximal amount of exploration possible with this filtering step. This baseline, "random10k", randomly selects a batch from the 10,000 candidates that are highest ranked by mean prediction.

### A.4.4 ANALYZING BATCH DIVERSITY

We visually analyze the diversity of batches selected by qPO, pTS, qEI, qPI, and UCB in Figure 2. We perform this analysis for batches selected in Iteration 1 of the run initialized with the random seed 7. At this iteration, all strategies have acquired the same training data and select from the same set of candidates. For each pair of selections in the batch, we calculate the Tanimoto similarity (Tanimoto, 1958) between 2048-length count Morgan fingerprints, leading to the histograms in the bottom of Figure 2. For network visualizations, each node represents a compound in the acquired batch. Edges are drawn between any pair of selections which have a Tanimoto similarity greater than 0.4. With each edge weight equal to the corresponding Tanimoto similarity, nodes are positioned with the Fruchterman-Reingold force-directed algorithm (Fruchterman & Reingold, 1991) as implemented in `networkx` (Hagberg et al., 2008) with a random seed of 0 (for reproducibility) and a $k$ value of

0.5 (to prevent highly overlapping nodes). This positioning algorithm visually clusters points that represent structurally similar compounds, allowing for qualitative analysis of batch diversity.

## A.5 RUNTIME COMPARISON BETWEEN qPO AND ALTERNATIVE STRATEGIES

| Method | Wall time for QM9 experiment (s) | Wall time for antibiotics experiment (s) |
|---|---|---|
| qPO | $3774 \pm 19$ | $1883 \pm 9$ |
| pTS | $3715 \pm 19$ | $1855 \pm 13$ |
| qEI | $6016 \pm 9$ | $1974 \pm 3$ |
| Greedy | $46 \pm 1$ | $20 \pm 0$ |
| UCB | $51 \pm 8$ | $26 \pm 8$ |
| TS-RSR | $3687 \pm 18$ | $1834 \pm 13$ |
| BUCB | $4840 \pm 12$ | $1401 \pm 6$ |
| qPI | $4850 \pm 12$ | $1385 \pm 6$ |
| DPP-TS | $5728 \pm 43$ | $2902 \pm 36$ |
| random10k | $44 \pm 2$ | $20 \pm 2$ |

Table 3: Wall times for qPO and baselines. Reported values denote the average wall time required for one complete run $\pm$ one standard error of the mean across ten runs. The runtime of qPO is within the same order of magnitude as that of other batch-level acquisition strategies that rely on sampling. As expected, strategies that do not require sampling–Greedy, UCB, and random10k–are significantly faster. All experiments were performed on nodes containing 40 CPUs and 2 GPUs.

### A.6 ABLATION STUDY ON IMPACT OF PRE-FILTERING THRESHOLD ON qPO PERFORMANCE

As described in Section 4 and A.4.3, we apply a pre-filtering before utilizing qPO and other sampling-based acquisition strategies to reduce the computational cost of these methods. For these strategies, we first select the top 10,000 points based on predicted mean and then apply the respective acquisition function. This may risk overlooking promising candidates; in particular, designs with poor predicted mean values but high uncertainty may fail to be considered for acquisition. Here, we compare performance of qPO using a greedy pre-filtering approach and one that instead uses upper confidence bound (UCB) to pre-filter compounds. In both cases, we maintain a threshold of 10,000 designs, and we perform this comparison for the application to antibiotic discovery described in Section 4.2. We observe similar optimization performance for qPO when using greedy and UCB metrics for pre-filtering according to the average of the top acquired compounds (Figure 4A,B), and slightly improved performance when using greedy according to retrieval of the top-performing compounds (Figure 4C,D). These results indicate that our pre-filtering method does not overlook promising designs with high uncertainty.

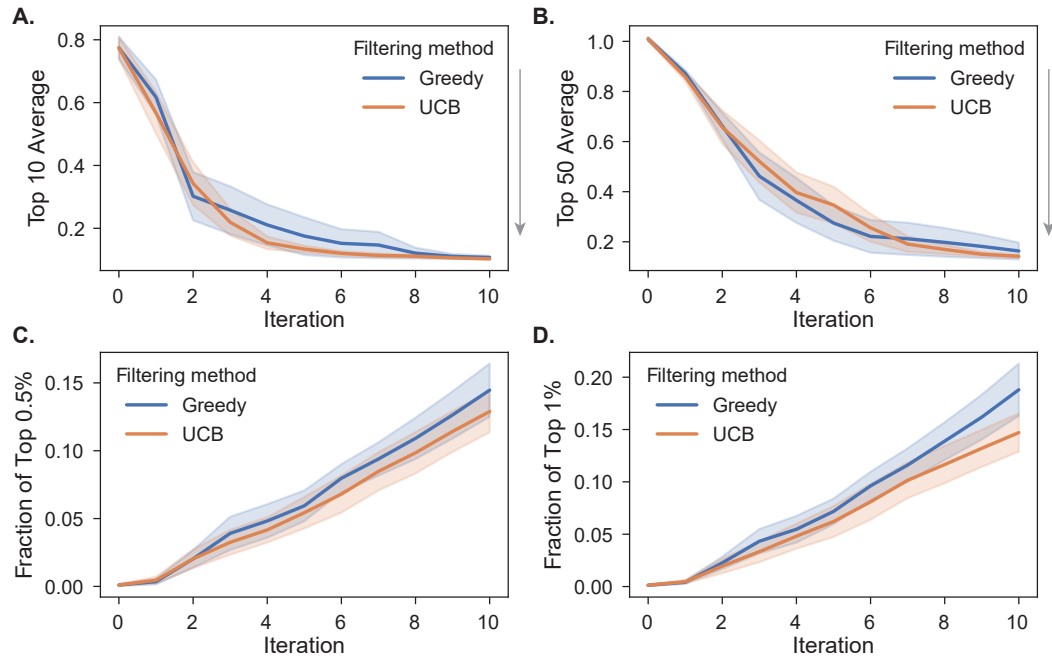

Figure 4: Comparison of qPO's optimization performance using greedy and upper confidence bound pre-filtering methods. Reported results are for the model-guided exploration of an experimental antibiotic activity dataset (Wong et al., 2024), as described in Section 4.2. In each iteration, the candidate set is filtered to 10,000 compounds before applying qPO acquisition, using either a greedy or upper confidence bound metric. (A,B) Average oracle value of top 10 and 50 acquired compounds, where lower values indicate greater antibiotic activity. (C, D) Retrieval of the true top 0.5% and 1% compounds in the explored library.

