# OpenReview forum: "Batched Bayesian optimization with correlated candidate uncertainties"
_ICLR.cc/2025/Conference — Submitted to ICLR 2025_

### Official Review · Reviewer_8R9v · 2024-10-24

**Soundness:** 2
**Presentation:** 2
**Contribution:** 1
**Rating:** 3
**Confidence:** 5

**Summary:**

This paper proposes a new batch acquisition function (qPO) for bayesian optimization in discrete search space. qPO is a purely exploitative approach, and its objective is to maximize the probability that the true optimum exists within the selected batch. qPO has nice property that it can be decomposed into a sum of scores over candidates, which allows naive parallelization in batch setting and simplifies the implementation.

The authors evaluate qPO on two molecular optimization tasks and show empirically that qPO performs comparably to or better than state-of-the-art batch acquisition method like q-TS, q-EI and UCB.

**Strengths:**

Originality
1. The formulation of batch acquisition function is novel, and its unique decomposition property where the batch acquisition function equals sum of individual probabilities is useful
2. Achieving batch diversity through formulation towards exploitation is creative

Quality
1. Mathematical formulation and derivation of qPO is clear and sound.

Clarity
1. Visualization of batch diversity in Figure 3 showing both network structure and similarity distributions is great.
2. The 3-point example in Section 3.2 helps illustrate key properties of the acquisition function

Significance
1. Introduces a simpler alternative to complex batch construction strategies
2. The decomposition property makes the method computationally tractable for moderated sized search space

**Weaknesses:**

1. There is no theoretical justification for why maximizing probability of optimality leads to better sample efficiency (compared with other existing batch acquisition functions like q-EI, q-PI, etc.), and convergence analysis is also absent
2. The diversity property is not unique to qPO as similar property exists in q-PI and other joint probability based methods, but the paper does not provide discussion for this
3. The empirical study is not sufficient:
    1. Missing standard BO benchmarks like synthetic functions
    2. No ablation studies to analyze impact of batch size, effect of pre-filtering threshold, sensitivity to number of MC samples, etc.
    3. There is no comparison to existing joint probability based methods like q-PI, as they are most similar to qPO
4. The pre-filtering strategy may miss promising candidates, therefore, there should be robustness analysis of this method

**Questions:**

1. Estimation of qPO through MC sampling seems computational expensive, can you provide runtime comparisons with other batch acquisition methods?
2. How robust is the method to model poor initial training data?
3. Does the pre-filtering step risk overlooking promising candidates in the regions with high uncertainty?

---

> ### Author Response · Authors · 2024-11-19
> **Authors' Response to Reviewer 8R9v (part 1 of 2)**
>
> We thank the reviewer for their thoughtful feedback on our submission. We have included our responses to the reviewer's comments and questions below. If there are any additional comparisons or clarifications in the text that might improve your impression and assessment of our work, please do not hesitate to let us know.
>
> **Weaknesses**
>
> >1. There is no theoretical justification for why maximizing probability of optimality leads to better sample efficiency (compared with other existing batch acquisition functions like q-EI, q-PI, etc.), and convergence analysis is also absent
>
> As mentioned in the response to reviewer PfuF, we agree that deriving convergence guarantees could support the use of qPO, however we think it is a mistake to view the absence of convergence guarantees as a flaw of qPO, because:
> 1. Convergence guarantees for other algorithms usually depend on idealized assumptions which may not hold in practice. For example, bounds derived for TS-RSR [2] and pTS [3] utilize bounds derived by Srinivas et al [4], which assume $f$ either is sampled from a known Gaussian process or has a bounded norm in a known RKHS.
> 2. Convergence guarantees are almost always upper bounds, which may not be tight
> 3. These bounds are usually quite loose for small numbers of iterations, which is the scenario targeted by qPO
> For this reason we focused mainly on experimental comparisons where we do observe benefits over qEI and qPI.
>
> > 2. The diversity property is not unique to qPO as similar property exists in q-PI and other joint probability based methods, but the paper does not provide discussion for this
>
> As the review mentions, qEI and qPI can encourage diversity through consideration of joint probability. In our evaluation of diversity in Fig. 3, we observe that qEI selects a less diverse batch than qPO, despite its implicit consideration of diversity. We have also now included qPI as a baseline and visualize the diversity of its selections in Figure 2.
>
> Additionally, in contrast to qPO, these acquisition functions cannot be reduced to a sum over individual acquisition scores and necessitate myopic batch construction, which may lead to the selection of suboptimal batches. qPO is able to capture diversity without myopic construction, a key distinction from qEI and qPI.
>
> > 3. The empirical study is not sufficient:
> >    1. Missing standard BO benchmarks like synthetic functions
> >    2. No ablation studies to analyze impact of batch size, effect of pre-filtering threshold, sensitivity to number of MC samples, etc.
> >    3. There is no comparison to existing joint probability based methods like q-PI, as they are most similar to qPO
>
> We are not aware of synthetic benchmarks designed for discrete optimization for sparsely populated high-dimensional design spaces, which is standard for molecular optimization tasks and the setting being addressed in this work. Synthetic benchmark functions (e.g., Ackley, Rosenbrock) are typically in low dimensions (<100), while molecules are featurized most commonly as discretized vectors 512-2048 in length. Therefore, we do not believe performance on continuous synthetic benchmark functions would indicate performance improvement for the application of interest: exploration of discrete molecular libraries.
>
> Among the baselines we have added is qPI. We have also added an ablation study comparing qPO optimization performance using our greedy pre-filtering strategy to one that uses an upper confidence bound metric.
>
>
> > 4. The pre-filtering strategy may miss promising candidates, therefore, there should be robustness analysis of this method
>
> Pre-filtering is not inherent to qPO. In our experiments, we employ pre-filtering as a matter of convenience to reduce the memory footprint of exact GP fitting in our experiments. We note that this type of filtering to reduce computational cost is not unprecedented; Moss et al [1, Section 6.4] apply a similar pre-filtering to 1,000 compounds for their model-guided chemical library screen demonstration of GIBBON .
>
> Nevertheless, it is of course possible that promising candidates are removed; it is possible in general that the surrogate model overlooks promising candidates in any model-guided optimization setting, which could deleteriously affect performance for any acquisition function.
>
> We now include an ablation study comparing qPO’s optimization performance using this greedy pre-filtering strategy to one that instead uses an upper confidence bound (UCB) metric. The UCB metric should recover candidates with high uncertainty that may be overlooked by a greedy pre-filtering strategy. The results of this study are now shown in Appendix A.6. qPO’s optimization performance is similar using both pre-filtering strategies, indicating that the our pre-filtering method does not miss a significant number of promising candidates.

---

> ### Author Response · Authors · 2024-11-19
> **Authors' Response to Reviewer 8R9v (part 2 of 2)**
>
> **Questions**
>
> > 1. Estimation of qPO through MC sampling seems computational expensive, can you provide runtime comparisons with other batch acquisition methods?
>
> We now report the average wall time for each method in the Appendix (A.5) and have copied it below. Overall, the wall time is within the same order of magnitude for all sampling-based acquisition functions.
>
> | Method    | Wall time for QM9 experiment (s) | Wall time for antibiotics experiment (s) |
> |-----------|----------------------------------|------------------------------------------|
> | qPO       | 3774 $\pm$ 19                    | 1883 $\pm$ 9                             |
> | pTS       | 3715 $\pm$ 19                    | 1855 $\pm$ 13                            |
> | qEI       | 6016 $\pm$ 9                     | 1974 $\pm$ 3                             |
> | Greedy    | 46 $\pm$ 1                       | 20 $\pm$ 0                               |
> | UCB       | 51 $\pm$ 8                       | 26 $\pm$ 8                               |
> | TS-RSR    | 3687 $\pm$ 18                    | 1834 $\pm$ 13                            |
> | BUCB      | 4840 $\pm$ 12                    | 1401 $\pm$ 6                             |
> | qPI       | 4850 $\pm$ 12                    | 1385 $\pm$ 6                             |
> | DPP-TS    | 5728 $\pm$ 43                    | 2902 $\pm$ 36                            |
> | random10k | 44 $\pm$ 2                       | 20 $\pm$ 2                               |
>
>
>
> > 2. How robust is the method to model poor initial training data?
>
> We have not performed any experiments at this time to explicitly probe this question. However, we would expect that prediction variances would be high when the initial training data is poor. In these cases, qPO scores would be more uniformly distributed across candidates. Therefore, qPO’s behavior would be considered exploratory under these conditions. This is in contrast to what would be expected at later iterations with sufficient and “good” training data, where variance would likely be lower and qPO scores would be very high for some candidates and close to zero for others. Despite this expectation, we do not yet have experimental results to answer this question with certainty. Investigating the sensitivity of optimization performance to poor initial training data through empirical evaluations is not a common practice or expectation in the field to our knowledge.
>
>
> > 3. Does the pre-filtering step risk overlooking promising candidates in the regions with high uncertainty?
>
> See our reply to your earlier comment.
>
>
> **Citations**
>
> [1] Moss, H. B.; Leslie, D. S.; González, J.; Rayson, P. GIBBON: General-Purpose Information-Based Bayesian Optimisation. J. Mach. Learn. Res. 2021, 22 (1), 235:10616-235:10664.
>
> [2] Ren, Z.; Li, N. TS-RSR: A Provably Efficient Approach for Batch Bayesian Optimization. arXiv 2024. arXiv:2403.04764
>
> [3] Kandasamy, K.; Krishnamurthy, A.; Schneider, J.; Poczos, B. Parallelised Bayesian Optimisation via Thompson Sampling. In Proceedings of the Twenty-First International Conference on Artificial Intelligence and Statistics; PMLR, 2018; pp 133–142.
>
> [4] Srinivas, N.; Krause, A.; Kakade, S.; Seeger, M. Gaussian Process Optimization in the Bandit Setting: No Regret and Experimental Design. In Proceedings of the 27th International Conference on International Conference on Machine Learning; ICML’10; Omnipress: Madison, WI, USA, 2010; pp 1015–1022.

---

### Official Review · Reviewer_gD5x · 2024-10-25

**Soundness:** 2
**Presentation:** 2
**Contribution:** 2
**Rating:** 3
**Confidence:** 3

**Summary:**

This paper investigates a novel acquisition function (qPO) for enhancing batch diversity in Thompson sampling. The approach shares conceptual similarities with entropy-search-based methods, as it leverages numerous Thompson sampling trials to estimate the posterior belief of the global optima, due to the non-closed-form nature of this distribution. The estimated function is then maximized to guide selection. While the proposed method relies on several approximation techniques to accelerate computation—whose statistical consistency and bias have yet to be thoroughly validated—it demonstrates superior performance over three baseline methods, which were not specifically designed for diversified batch sampling.

**Strengths:**

- Highly parallel Bayesian Optimization is an important challenge in real-world tasks, though numerous existing works have pursued similar approaches with clearer conceptual frameworks and stronger theoretical foundations.
- While real-world examples add interest, they rely solely on existing datasets, such as QM9, and thus do not introduce new applications. This work remains primarily an algorithm-focused paper.
- While I appreciate the authors’ effort in presenting Figure 1 to convey the intuition behind the method, I would have liked to see a more precise mathematical explanation.

**Weaknesses:**

My primary reasons for recommending rejection are as follows: (1) issues with clarity, particularly in the mathematical exposition, (2) limited insights and contributions to the community, and (3) an unfair selection of baselines, which undermines the validity of the claimed empirical improvements.

- **Clarity**
One of the major issues in this paper is clarity, particularly regarding the mathematical exposition. I had a hard time to understand Eqs. (6)–(11). My detailed concerns are as follows:

**Eqs. (6)–(10):** What do the bold vectors $\textbf{x}$ and $\textbf{y}$ represent? Are they the observed points so far, or do they represent all candidates, where $|\textbf{x}| = N$? The former interpretation doesn’t make much sense, so I assume it’s the latter. In that case, this expression becomes a utility function for probability of improvement (PI) (see page 2 in [1]). Then $p(\textbf{y} \mid \textbf{x})$ in Eq. (9) would describe an ideal scenario where we have access to the entire domain $\textbf{x}$, $\textbf{y}$. However, this understanding conflicts with the textual explanations. Specifically, in lines 192–193, the reference to “myopic construction” isn’t clear. If this is indeed the PI utility function $u$, then it would follow that the resulting PI is myopic. Although $Pr(x^\star)$ is indeed the optimal acquisition function, it’s also not achievable in practice, since we could identify $x^\star$ in a single iteration if we had access to the “oracle acquisition function.” Thus, we use a surrogate model to estimate $x^\star$, resulting in an estimate $\hat{x}^\star$, not $x^\star$. Consequently, the myopicness originates from the belief model $f$, not the utility function $u$. Thus, the entire discussion is confusing.

**Eq. (11):** This equation is especially difficult to interpret. What do $y^{(m)}_i$ and $\textbf{y}^{(m)}$ represent? I assume $\textbf{y}^{(m)}$ is a sample function drawn from the GP posterior, with $\max \textbf{y}^{(m)}$ as its maximum, similar to Thompson sampling (TS). However, what is $y_i$? Is it the predictive mean $m(x_i)$? If my understanding is correct, then only one point should be true, with all others zeroed out in the indicator count $\textbf{1}$. Isn’t this an embarassingly inefficient Monte Carlo estimation? Even if we were to take a very large number of samples $M$, the belief model is not the true function. This would imply that the resulting estimate is the posterior belief of the optimum $Pr(\hat{x}^* = x_i)$. Why, then, would this recover the probability of the true optimum $x^*$? This is not strictly an approximation of Eq. (10); while it may be asymptotically convergent, further justification is required to establish this.

- **Unclear Benefits and Inaccurate Descriptions**
Due to clarity issues, I was unable to fully understand how this method differs from existing approaches, such as entropy-search-based methods [2] or other principled approaches in diversified parallel TS. First, the term "correlated candidate uncertainty" in the title is unclear, as it isn’t fully explained within the paper. If this refers to the diversification of the batch selection, there are many existing works. On line 320, the phrase “Deterministically maximize” is somewhat misleading. This statement would hold if Eq. (11) had a closed-form solution. However, since this algorithm relies on multiple approximations, it will not be deterministic due to numerical randomness, similar to TS. Additionally, the “diversity heuristics” mentioned on line 280 are inaccurately described. For instance, Nava et al. (2022) demonstrated a theoretical improvement in convergence rates over parallel TS, showing that these are not merely heuristics but instead clearly proven benefits. In contrast, this paper does not present any convergence analysis results, making it a heuristic approach. Could you clarify which aspects of the parallel TS convergence rate [3] qPO improves upon? Is such improvement better than recent work by Nava et al. (2022) and Ren & Li (2024), which proves tighter regret bounds via batch diversity? If this method (qPO) cannot achieve theoretical or empirical improvements over these methods, I see limited justification for adopting it over these established approaches.

- **Robustness Issues**
Recent work [4,5] raises issues that contrast with the claims in this paper. While this paper argues that parallel TS is overly explorative and suggests an exploitative approach as a remedy, [4,5] contend that parallel TS lacks sufficient diversity and instead requires an explorative adjustment. Intuitively, one would expect an exploitative approach to reduce batch diversity rather than enhance it. Why, then, does this method claim that exploitation leads to diversification? Additionally, [4] and [5] underscore robustness issues in parallel TS-based methods, specifically regarding accurate sample drawing and model misspecification. For example, [4] shows that heuristic approximations, such as the greedy sampling used in this paper, introduce bias in approximating the probability distribution, proposing Matheron’s rule as a corrective approach. Furthermore, [5] demonstrates that if GP hyperparameters are misspecified (e.g., lengthscale), the Monte Carlo estimate in TS becomes biased, and they address this with a robust kernel quadrature approach. Both of these concerns are relevant to the tasks considered in this paper: the sampling approach does not utilize an exact Cholesky decomposition, and the Tanimoto kernel lacks a consistency guarantee for MLE-based hyperparameter estimation. More generally, hyperparameter estimation with MLE in GPs is theoretically ill-posed [8]. Since this paper does not address these robustness concerns, it is unclear what progress or insights it contributes, especially since both [4] and [5] achieve improved diversity akin to the results here.

- **Unfair Baselines**. It’s also worth noting that the baselines selected in this paper are outdated and do not reflect recent advancements. For instance, the qEI baseline should employ sample average approximation with quasi-Monte Carlo as in the original BoTorch paper [6], while UCB should ideally use DB-GP-UCB [7], which is a more theoretically grounded batch approach in UCB. Furthermore, for pTS, which is the main target of this work, a fair comparison would require recent theoretical contributions, such as those by Nava et al. (2022) and/or Ren & Li (2024), as well as empirical work from [4, 5]. Without these comparisons, it is difficult to gauge the practical utility of this method. Given the variety of heuristic and theoretical methods already available, if this approach is intended to be an improvement, a comprehensive empirical comparison with the aforementioned methods is essential. Alternatively, if the goal is to provide new insights, the paper should offer theoretical analysis and/or a clearer empirical study to identify scenarios where this method excels yet other state-of-the-art methods fail, making it more relevant to the community. While this method may have presented an improvement over parallel TS in 2017, it now appears outdated.

- **Minor points**. I also believe that [3] should be cited as a foundational reference for parallel TS, as it provides a clearer theoretical understanding of the method. While Hernández-Lobato et al. (2017) originally proposed it, [3] now serves as a key reference for the theoretical insights underpinning parallel TS.

- **Citations**
- [1] https://www.cse.wustl.edu/~garnett/cse515t/spring_2015/files/lecture_notes/12.pdf
- [2] Takeno, Shion et al., "Sequential and Parallel Constrained Max-value Entropy Search via Information Lower Bound", ICML 2023
- [3] Kandasamy, Kirthevasan, et al. "Parallelised Bayesian optimisation via Thompson sampling." AISTATS 2018
- [4] Wilson, James, et al. "Efficiently sampling functions from Gaussian process posteriors." *ICML 2020*
- [5] Adachi, Masaki, et al. "A Quadrature Approach for General-Purpose Batch Bayesian Optimization via Probabilistic Lifting." arXiv:2404.12219 (2024).
- [6] Balandat, Maximilian, et al. "BoTorch: A framework for efficient Monte-Carlo Bayesian optimization." NeurIPS (2020).
- [7] Daxberger, E. A. et al. "Distributed batch Gaussian process optimization". ICML (2017)
- [8] Karvonen, T. et al. "Maximum Likelihood Estimation in Gaussian Process Regression is Ill-Posed". JMLR (2023)

**Questions:**

See the weakness section.

---

> ### Author Response · Authors · 2024-11-19
> **Authors' Response to Reviewer gD5x (part 1 of 4)**
>
> We thank the reviewer for their thoughtful feedback on our submission. We have included our responses to the reviewer's comments and questions below. If there are any additional comparisons or clarifications in the text that might improve your impression and assessment of our work, please do not hesitate to let us know.
>
> **Weaknesses**
>
> > My primary reasons for recommending rejection are as follows: (1) issues with clarity, particularly in the mathematical exposition, (2) limited insights and contributions to the community, and (3) an unfair selection of baselines, which undermines the validity of the claimed empirical improvements.
>
> We thank the reviewer for highlighting these weaknesses and have responded to specific comments below.
>
> > **Clarity**
> >
> > One of the major issues in this paper is clarity, particularly regarding the mathematical exposition. I had a hard time to understand Eqs. (6)–(11). My detailed concerns are as follows:
> >
> > Eqs. (6)–(10): What do the bold vectors $\textbf{x}$ and $\textbf{y}$ represent? Are they the observed points so far, or do they represent all candidates, where $|\textbf{x}|=N$? The former interpretation doesn’t make much sense, so I assume it’s the latter.
>
> The reviewer is correct; $\textbf{x}$ represents the set of candidates from which a batch is selected. We thank the reviewer for pointing out our inconsistency in the definition of $f$ and $y$. $f$ is defined as the ground truth objective value, while $y$ is now exclusively defined as surrogate model predictions of $f$. We have modified the text in Section 2.2 and in Algorithm 1 to convey this more clearly.
>
> > In that case, this expression becomes a utility function for probability of improvement (PI) (see page 2 in [1]).
>
> For a batch size of 1, PI considers the improvement in objective score over that of previously acquired points. In the same scenario, qPO considers a candidates’ superiority over all other (unacquired) candidates, which is distinct from previously acquired points. This distinction also applies to the batched extension. Therefore, this expression is not equivalent to the PI utility function.
>
> > Then $p(\textbf{y} | \textbf{x})$ in Eq. (9) would describe an ideal scenario where we have access to the entire domain $\textbf{x}$,$\textbf{y}$.
>
> Equation 9 involves the integral over $\textbf{y}$, which represents the surrogate model posterior predictions for the candidates in $\textbf{x}$. We do not assume access to the entire domain’s ground truth objective value, but we assume knowledge of the entire set of candidates $\textbf{x}$ that define the finite discrete design space we are optimizing within. We understand that this was not clear in our initial submission.
>
> > However, this understanding conflicts with the textual explanations. Specifically, in lines 192–193, the reference to “myopic construction” isn’t clear. If this is indeed the PI utility function $u$, then it would follow that the resulting PI is myopic. Although $Pr(x^*)$ is indeed the optimal acquisition function, it’s also not achievable in practice, since we could identify $x^*$ in a single iteration if we had access to the “oracle acquisition function.” Thus, we use a surrogate model to estimate $x^*$, resulting in an estimate $\hat{x}^*$, not $x^*$. Consequently, the myopicness originates from the belief model $f$, not the utility function $u$. Thus, the entire discussion is confusing.
>
> It is correct that we use a surrogate model to estimate the probability of each point being the optimum from the set of candidates. qPI is myopic because its batch-level acquisition function cannot be formulated as a sum of individual acquisition scores, making its optimization a combinatorial problem. qPO, which has a different utility function, can be reduced to a sum over individual acquisition scores if certain assumptions can be made about the surrogate model. We have clarified this discussion and thank you for pointing out the source of confusion.
>
> > Eq. (11): This equation is especially difficult to interpret. What do $y_i^{(m)}$ and $\textbf{y}_i^{(m)}$ represent? I assume $\textbf{y}_i^{(m)}$ is a sample function drawn from the GP posterior, with $\text{max } \textbf{y}^{(m)}$ as its maximum, similar to Thompson sampling (TS).
>
> This is correct. We have clarified this in the text following the referenced Equation.
>
> > However, what is $y_i$? Is it the predictive mean $m(x_i)$? If my understanding is correct, then only one point should be true, with all others zeroed out in the indicator count $\bf{1}$. Isn’t this an embarassingly inefficient Monte Carlo estimation?
>
> We have clarified our definition of $y_i$ in the text as the surrogate model prediction of $f_i$, which is a random variable following the surrogate model posterior. Therefore, the value of $y_i$ cannot be described by only one point.

---

> ### Author Response · Authors · 2024-11-19
> **Authors' Response to Reviewer gD5x (part 2 of 4)**
>
> >**Clarity** (continued)
> >
> > Even if we were to take a very large number of samples $M$, the belief model is not the true function. This would imply that the resulting estimate is the posterior belief of the optimum $Pr(\hat{x}^*=x_i)$.
>
> This is correct. It would not be appropriate to assume access to the true function. We have updated the text and notation in Section 2 to better convey this point. qPO selects points that optimize $Pr(x^*=x_i)$ according to the surrogate model, that is $Pr(\hat{x}^*=x_i)$ according to the reviewer.
>
> > Why, then, would this recover the probability of the true optimum $x^*$? This is not strictly an approximation of Eq. (10); while it may be asymptotically convergent, further justification is required to establish this.
>
> qPO uses the surrogate model to estimate the likelihood of a candidate being the true optimum, similar to how qPI uses the surrogate model to estimate the likelihood of a candidate having an objective value greater than the current best.
>
>
> > **Unclear Benefits and Inaccurate Descriptions**
> >
> > Due to clarity issues, I was unable to fully understand how this method differs from existing approaches, such as entropy-search-based methods [2] or other principled approaches in diversified parallel TS. First, the term "correlated candidate uncertainty" in the title is unclear, as it isn’t fully explained within the paper. If this refers to the diversification of the batch selection, there are many existing works.
>
> We have updated our title to “Batched Bayesian optimization in discrete domains by maximizing the probability of including the optimum”. We believe this title is more clear and representative of the work than our previous title.
>
> > On line 320, the phrase “Deterministically maximize” is somewhat misleading. This statement would hold if Eq. (11) had a closed-form solution. However, since this algorithm relies on multiple approximations, it will not be deterministic due to numerical randomness, similar to TS.
>
> As the reviewer describes, qPO cannot be deterministically optimized because there is no closed form solution to qPO acquisition scores. However, the aim of qPO is still to select designs deterministically, as this would occur in the case $M \rightarrow \infty$. This contrasts pTS, which intentionally selects points randomly. The distinction we meant to draw is whether determinism is intended by the design of the acquisition function, not whether it is achieved by a numerical implementation of it. We have modified the phrasing of Section 3.2 to clarify this important point.
>
> > Additionally, the “diversity heuristics” mentioned on line 280 are inaccurately described. For instance, Nava et al. (2022) demonstrated a theoretical improvement in convergence rates over parallel TS, showing that these are not merely heuristics but instead clearly proven benefits. In contrast, this paper does not present any convergence analysis results, making it a heuristic approach.
>
> We have updated our description of Nava et al. (2022) [1] in the text (Section 3.1) and also included this work as a baseline. (Please see the next comment/reply for additional information addressing the other part of this comment)
>
> > Could you clarify which aspects of the parallel TS convergence rate [3] qPO improves upon? Is such improvement better than recent work by Nava et al. (2022) and Ren & Li (2024), which proves tighter regret bounds via batch diversity? If this method (qPO) cannot achieve theoretical or empirical improvements over these methods, I see limited justification for adopting it over these established approaches.
>
> We have included Nava et al. (2022) [1] and Ren & Li (2024) [2] as baselines in our updated submission.
>
> As mentioned in the response to reviewer PfuF, we agree that deriving convergence guarantees could support the use of qPO, however we think it is a mistake to view the absence of convergence guarantees as a flaw of qPO, because:
> 1. Convergence guarantees for other algorithms usually depend on idealized assumptions which may not hold in practice. For example, bounds derived for TS-RSR [2] and pTS [6] utilize bounds derived by Srinivas et al [7], which assume $f$ either is sampled from a known Gaussian process or has a bounded norm in a known RKHS.
> 2. Convergence guarantees are almost always upper bounds, which may not be tight
> 3. These bounds are usually quite loose for small numbers of iterations, which is the scenario targeted by qPO
> For this reason we focused mainly on experimental comparisons where we do observe benefits.
>
> Despite the exclusion of convergence rate analysis, we would like to respectfully disagree that qPO is merely a heuristic approach; the acquisition function is defined by a clear and defensible statement of optimality at each iteration, though we recognize (as you have pointed out earlier) that a surrogate model is used to estimate likelihoods that cannot be known exactly without access to the ground truth objective function.

---

> ### Author Response · Authors · 2024-11-19
> **Authors' Response to Reviewer gD5x (part 3 of 4)**
>
> Authors' Response to Reviewer gD5x (part 3 of 4)
>
> > **Robustness Issues**
> >
> > Recent work [4,5] raises issues that contrast with the claims in this paper. While this paper argues that parallel TS is overly explorative and suggests an exploitative approach as a remedy, [4,5] contend that parallel TS lacks sufficient diversity and instead requires an explorative adjustment.
>
> Section 3.2 describes an example of how pTS may fail to select a diverse batch, which is consistent with the observations referenced by the reviewer. Further, this example demonstrates how qPO promotes selection of a diverse batch. We do not intend to imply that parallel TS is overly explorative, only that qPO may lead to improved performance relative to pTS in the application of interest, which we demonstrate through empirical results.
>
> > Intuitively, one would expect an exploitative approach to reduce batch diversity rather than enhance it. Why, then, does this method claim that exploitation leads to diversification?
>
> Exploitation and exploration are broadly defined concepts, and we understand that qPO’s preference for diverse batches may be counterintuitive. We argue that the motivation of maximizing the probability of acquiring the true optimum is an exploitative goal. However, we demonstrate in Section 3.2 and Figure 3 how this acquisition function achieves greater diversity than other exploitative methods (e.g. UCB). While this may seem unintuitive, one might alternatively interpret this observation as qPO incorporating some element of exploration despite its exploitative motivation. We have modified our introduction and discussion of qPO throughout the text to highlight that while the motivation for its derivation was exploitative, the acquisition strategy does exhibit exploratory behavior, specifically in its selection of diverse batches.
>
> > Additionally, [4] and [5] underscore robustness issues in parallel TS-based methods, specifically regarding accurate sample drawing and model misspecification. For example, [4] shows that heuristic approximations, such as the greedy sampling used in this paper, introduce bias in approximating the probability distribution, proposing Matheron’s rule as a corrective approach. Furthermore, [5] demonstrates that if GP hyperparameters are misspecified (e.g., lengthscale), the Monte Carlo estimate in TS becomes biased, and they address this with a robust kernel quadrature approach. Both of these concerns are relevant to the tasks considered in this paper: the sampling approach does not utilize an exact Cholesky decomposition, and the Tanimoto kernel lacks a consistency guarantee for MLE-based hyperparameter estimation. More generally, hyperparameter estimation with MLE in GPs is theoretically ill-posed [8]. Since this paper does not address these robustness concerns, it is unclear what progress or insights it contributes, especially since both [4] and [5] achieve improved diversity akin to the results here.
>
> While there are certainly some theoretical issues with Gaussian process surrogate model-fitting and additional ones related to the Tanimoto in particular, this surrogate model architecture has empirically outperformed alternatives for molecular optimization applications [5]. In the application of BO methods, theoretical guarantees must be balanced with empirical results, which may limit the use of the most robust algorithms without bias and with theoretical guarantees.

---

> ### Author Response · Authors · 2024-11-19
> **Authors' Response to Reviewer gD5x (part 4 of 4)**
>
> > **Unfair Baselines**
> >
> > It’s also worth noting that the baselines selected in this paper are outdated and do not reflect recent advancements. For instance, the qEI baseline should employ sample average approximation with quasi-Monte Carlo as in the original BoTorch paper [6], while UCB should ideally use DB-GP-UCB [7], which is a more theoretically grounded batch approach in UCB. Furthermore, for pTS, which is the main target of this work, a fair comparison would require recent theoretical contributions, such as those by Nava et al. (2022) and/or Ren & Li (2024), as well as empirical work from [4, 5]. Without these comparisons, it is difficult to gauge the practical utility of this method. Given the variety of heuristic and theoretical methods already available, if this approach is intended to be an improvement, a comprehensive empirical comparison with the aforementioned methods is essential. Alternatively, if the goal is to provide new insights, the paper should offer theoretical analysis and/or a clearer empirical study to identify scenarios where this method excels yet other state-of-the-art methods fail, making it more relevant to the community. While this method may have presented an improvement over parallel TS in 2017, it now appears outdated.
>
> In this revision, we have included the following baselines: qPI, batch UCB (as described in [3]), GIBBON [4], TS-RSR [2], and DPP-TS [1]. We hope that their inclusion increases your confidence in the strong empirical results we show for qPO. In general, we observe that qPO performs similarly to or better than DPP-TS [1] and TS-RSR [2] in both experiments.
>
> If there are additional methods beyond the ones you have listed here that you would recommend including, we will be glad to do so in the revised version of the manuscript.
>
> > **Minor points**
> >
> > I also believe that [3] should be cited as a foundational reference for parallel TS, as it provides a clearer theoretical understanding of the method. While Hernández-Lobato et al. (2017) originally proposed it, [3] now serves as a key reference for the theoretical insights underpinning parallel TS.
>
> We have revised our manuscript to include this key reference.
>
> **Citations**
>
> [1] Nava, E.; Mutny, M.; Krause, A. Diversified Sampling for Batched Bayesian Optimization with Determinantal Point Processes. In Proceedings of The 25th International Conference on Artificial Intelligence and Statistics; PMLR, 2022; pp 7031–7054.
>
> [2] Ren, Z.; Li, N. TS-RSR: A Provably Efficient Approach for Batch Bayesian Optimization. arXiv 2024. arXiv:2403.04764.
>
> [3] Wilson, J. T.; Moriconi, R.; Hutter, F.; Deisenroth, M. P. The Reparameterization Trick for Acquisition Functions. arXiv. 2017. arXiv:1712.00424.
>
> [4] Moss, H. B.; Leslie, D. S.; González, J.; Rayson, P. GIBBON: General-Purpose Information-Based Bayesian Optimisation. J. Mach. Learn. Res. 2021, 22 (1), 235:10616-235:10664.
>
> [5] Griffiths, R.-R.; Klarner, L.; Moss, H.; Ravuri, A.; Truong, S. T.; Du, Y.; Stanton, S. D.; Tom, G.; Ranković, B.; Jamasb, A. R.; Deshwal, A.; Schwartz, J.; Tripp, A.; Kell, G.; Frieder, S.; Bourached, A.; Chan, A. J.; Moss, J.; Guo, C.; Dürholt, J. P.; Chaurasia, S.; Park, J. W.; Strieth-Kalthoff, F.; Lee, A.; Cheng, B.; Aspuru-Guzik, A.; Schwaller, P.; Tang, J. GAUCHE: A Library for Gaussian Processes in Chemistry. In Thirty-seventh Conference on Neural Information Processing Systems; 2023.
>
> [6] Kandasamy, K.; Krishnamurthy, A.; Schneider, J.; Poczos, B. Parallelised Bayesian Optimisation via Thompson Sampling. In Proceedings of the Twenty-First International Conference on Artificial Intelligence and Statistics; PMLR, 2018; pp 133–142.
>
> [7] Srinivas, N.; Krause, A.; Kakade, S.; Seeger, M. Gaussian Process Optimization in the Bandit Setting: No Regret and Experimental Design. In Proceedings of the 27th International Conference on International Conference on Machine Learning; ICML’10; Omnipress: Madison, WI, USA, 2010; pp 1015–1022.

---

> > ### Comment · Reviewer_gD5x · 2024-11-25
> >
> > Thank you for your response. To summarize: (a) explanations remain unclear, (b) insights are still unclear, and (c) while the baselines have been improved, the work still omits comparisons to [4, 5, 7]. As a result, I will maintain my scores. Below are the detailed comments:
> >
> > ### Clarity
> > While some parts of your response are clearer, several non-trivial points remain difficult to follow, and some rebuttals appear incorrect. For example:
> > - I still find Eqs. (6)–(9) unclear.
> > - The justification for taking subsamples from the entire domain is insufficient.
> > - The proposed qPO method appears myopic, as it does not consider the rollout for future iterations, unlike typical non-myopic BO approaches with dynamic programming.
> > - The interpretation of $Pr(\hat{x} = x^*)$ is problematic. If it were a likelihood, the function value would not change over time. For comparison, consider the Gaussian likelihood of $y$, which remains consistent across iterations. Likelihood is a metric, so this appears closer to a reward function (i.e., an expected utility function). The probability of improvement does not treat $Pr(f(x) \geq \max{y})$ as a likelihood; it functions as an expected utility. This distinction needs to be clarified.
> >
> > ### Insights
> > Regarding theory, while I acknowledge the limitations of regret-based analyses, this does not justify omitting an explanation for why the proposed method performs well as a global optimizer. Providing such insights is critical for scientific contribution.
> >
> > - The strong assumptions referenced in the rebuttal are shared by your method. If the target function is not sampled from your GP model, how can $Pr(\hat{x}^*)$ be estimated using Thompson sampling? This also requires the additional assumption that GP hyperparameters are known. At this stage, these assumptions are unavoidable and are not unique to theoretical work.
> > - Recent work has progressed beyond requiring RKHS bounds (e.g., [1]). Furthermore, while convergence rates are typically presented as upper bounds, lower bounds are also becoming common (e.g., [2]).
> > - The issue with loose bounds on short horizons is a good point. This suggests a focus on constants rather than just the order of convergence rates. This point makes sense to me, especially since the Tanimoto kernel's spectral decay is notably slower than that of popular kernels like RBF on typical test functions. Since GP-UCB and TS depend on spectral decay, it makes sense that it performs poorly with the Tanimoto kernel. However, this raises the question: how does qPO reduce these constants? Exploring factors contributing to early-stage acceleration could clarify many unclear aspects of the submission, including comparisons with the conceptual pros and cons of existing methods.
> >
> > ### Experimental Validation
> > Section 3.2 illustrates the difference between parallel TS and qPO, but it does not provide comparisons with other approaches. Diversified batch TS is not a novel topic, and readers will expect a discussion of the issues with existing methods and how the proposed solution addresses them. Clearer explanations on this point are necessary. I think investigating the constant in the convergence rate is a promising direction, as it could strengthen the experimental narrative.
> >
> > Good luck with your next resubmission and hope my review helps to improve your work.
> >
> > [1] Felix Berkenkamp et al., "No-Regret Bayesian Optimization with Unknown Hyperparameters" JMLR 2019
> > [2] Scarlett, Jonathan, et al., "Lower Bounds on Regret for Noisy Gaussian Process Bandit Optimization", COLT 2017

---

### Official Review · Reviewer_tsqA · 2024-11-01

**Soundness:** 3
**Presentation:** 3
**Contribution:** 2
**Rating:** 5
**Confidence:** 4

**Summary:**

This paper proposes a batch acquisition function for Bayesian optimization on finite (and necessarily discrete) domains.
The acquisition function is the probability that the selected candidates contain the global optimum:
\\[
    \Pr(x^* \in \mathcal{X} _ {\text{selected candidates}}).
\\]
While it's quite challenging to compute this probability on continuous domains, it's easier to estimate this probability on discrete domains.
Indeed, the authors propose approximating this probability using Monte Carlo samples.
On antibiotic discovery and organic electronics design problems, the authors show that the proposed method outperform parallel Thomson sampling, multiple point EI, and UCB.

**Strengths:**

- The method is intuitive and simple.
- The writing is overall good.
- The improvement over parallel Thompson sampling, qEI, and UCB seems to be consistent across several datasets.

**Weaknesses:**

- This paper focuses on finite domains, which certainly simplifies the problem a lot.
This is reasonable.
However, if the primary application is molecule designs, I'd like to see comparisons with recent alternative Bayesian optimization methods that do not frame this problem as a finite discrete BO problem.

There are several immediate ideas worth considering. Without trying these ideas, the paper seems underdeveloped at this moment.
1. As noted by the authors, the probability \\(\Pr(x^* = x_i)\\) is an orthant normal probability, which can be handled by many existing Monte Carlo methods.
These methods are probably much more efficient and accurate than the proposed estimation method in this paper.
In particular, the method of Genz (1992), which is based on importance sampling, is not too hard to implement.
So employing better numerical methods estimating this normal probability is a nice plus.
2. The biggest improvement actually comes from filtering \\(10000\\) candidates using a greedy metric (which seems to be the posterior mean).
The authors argue that they do so due to the large computational cost of GP posterior sampling.
However, the experiments in Section 4.2 (\\(n = 39,312\\)) should be easily handle by modern GP packages, and the experiments in Section 4.3 (\\(n = 133,000\\)) should be easily handled by variational GPs.

- Also, in my opinion, the paper title is not reflective of the paper content.

**Questions:**

- The authors mentioned multiple times that the proposed acquisition function is purely exploitative.
I am not sure if this statement is accurate.
On a high level, the method seeks candidates that are more likely to be the optimum.
Its spirit is similar to predictive entropy search in the continuous setting, which is not purely exploitative.

- While the writing is good overall, it's not clear what the candidates \\(\\mathbf{x}\\) are at line 180. (Also at Line 220 ALgorithm 1).
Based on the experiments, it seems that \\(\mathbf{x}\\) is a subset of the domain selected by the greedy strategy (based on posterior mean)?

- I am not sure how Equation (4) makes sense.

---

> ### Author Response · Authors · 2024-11-19
> **Authors' Response to Reviewer tsqA (part 1 of 2)**
>
> We thank the reviewer for their thoughtful feedback on our submission. We have included our responses to the reviewer's comments and questions below. If there are any additional comparisons or clarifications in the text that might improve your impression and assessment of our work, please do not hesitate to let us know.
>
> **Weaknesses**
>
> >This paper focuses on finite domains, which certainly simplifies the problem a lot. This is reasonable. However, if the primary application is molecule designs, I'd like to see comparisons with recent alternative Bayesian optimization methods that do not frame this problem as a finite discrete BO problem.
>
> We focus on finite domains to address the application of interest: exploration of chemical libraries, which are finite discrete domains. [references related to this topic in ML venues] Therefore, we deem it appropriate to treat this problem as a finite discrete BO problem and compare with relevant methods. Nonetheless, we have incorporated additional baselines into our evaluation, including some that are designed for continuous domains. These include qPI, GIBBON [1], DPP-TS [2], and batch UCB [3]. We ask that the reviewer please clarify which other methods they deem appropriate for the proposed comparison if this list is not satisfactory, and we will be glad to include them in our final empirical evaluation.
>
> >There are several immediate ideas worth considering. Without trying these ideas, the paper seems underdeveloped at this moment.
>
> Thank you for bringing these points to our attention.
>
> >1. As noted by the authors, the probability $\Pr(x^*=x_i)$ is an orthant normal probability, which can be handled by many existing Monte Carlo methods. These methods are probably much more efficient and accurate than the proposed estimation method in this paper. In particular, the method of Genz (1992), which is based on importance sampling, is not too hard to implement. So employing better numerical methods estimating this normal probability is a nice plus.
>
> In our updated codebase, we include an implementation of our acquisition strategy by estimating orthant probabilities using Genz [4]. For our demonstrations in Section 4.2, we have tried implementing this and observe this method to be significantly more computationally expensive. We can justify this observation with the following complexity analysis: the proposed estimation requires sampling from a single multivariate distribution, requiring Cholesky decomposition of an $N \times N$ covariance matrix (for $N$ candidates) to be performed once. Using [4], estimating acquisition scores as orthants requires Cholesky decomposition to be evaluated for $N$ distinct $N-1 \times N-1$ covariance matrices, as each of $N$ candidates requires the estimation of a different orthant probability. Therefore, the scaling of our method is $O(N^3)$, while the scaling of using Genz’s method [4] is $O(N^4)$.
>
> > 2. The biggest improvement actually comes from filtering 10,000 candidates using a greedy metric (which seems to be the posterior mean). The authors argue that they do so due to the large computational cost of GP posterior sampling. However, the experiments in Section 4.2 ($n=39,312$) should be easily handle by modern GP packages, and the experiments in Section 4.3 ($n=133,000$) should be easily handled by variational GPs.
>
> If this statement were correct – that the biggest improvement comes from filtering 10,000 candidates – we would expect the random_10k baseline to perform similarly to pTS and qPO. However, we observe that the random_10k baseline performs poorly compared to other methods, indicating that filtering is not the source of the greatest improvement. We have based our implementations on BoTorch, where the memory footprint is limiting at these large candidate sizes, but agree that other approaches like variational GPs provide a path to address the larger candidate space, outside the scope of this work. We also note that this type of filtering due to computational cost is not unprecedented; Moss et al [1, Section 6.4] apply a similar pre-filtering to 1,000 compounds for their model-guided chemical library screen demonstration of GIBBON.
>
> > Also, in my opinion, the paper title is not reflective of the paper content.
>
> Thank you for the suggestion. We have revised the paper title to “Batched Bayesian optimization in discrete domains by maximizing the probability of including the optimum”

---

> ### Author Response · Authors · 2024-11-19
> **Authors' Response to Reviewer tsqA (part 2 of 2)**
>
> **Questions**
> > The authors mentioned multiple times that the proposed acquisition function is purely exploitative. I am not sure if this statement is accurate. On a high level, the method seeks candidates that are more likely to be the optimum. Its spirit is similar to predictive entropy search in the continuous setting, which is not purely exploitative.
>
> We believe that the goal of selecting points most likely to be optimal is conceptually an exploitative one. However, as the reviewer implies, because qPO promotes diversity, it may be interpreted as not purely exploitative. We modify our introduction and discussion of qPO throughout the text appropriately, highlighting that it was *motivated* by exploitation but exhibits exploratory behavior (e.g. by encouraging diversity).
>
> > While the writing is good overall, it's not clear what the candidates $\textbf{x}$ are at line 180. (Also at Line 220 ALgorithm 1). Based on the experiments, it seems that $\textbf{x}$ is a subset of the domain selected by the greedy strategy (based on posterior mean)?
>
> $\textbf{x}=[x_1, x_2, …]$ represents a set of candidates from which a batch is selected. In Algorithm 1, it refers to the subset of the domain which has not already been acquired in previous BO iterations. We have updated the text in Section 2.2 and in Algorithm 1 to clarify this point. For our experiments in Section 4 specifically, the reviewer is correct; $\textbf{x}$ is the subset of the domain that are candidate inputs to qPO, which is a modification to Algorithm 1. We have updated Section 4.1 accordingly. It is important to note that qPO itself does not require this greedy pre-filtering step in general, which is why Algorithm 1 does not include it.
>
> > I am not sure how Equation (4) makes sense.
>
> Thank you for bringing this to our attention. After reassessing our derivation, we have realized that this equation is not necessary and have removed it from our most recent revision.
>
> **Citations**
>
> [1] Moss, H. B.; Leslie, D. S.; González, J.; Rayson, P. GIBBON: General-Purpose Information-Based Bayesian Optimisation. J. Mach. Learn. Res. 2021, 22 (1), 235:10616-235:10664.
>
> [2] Nava, E.; Mutny, M.; Krause, A. Diversified Sampling for Batched Bayesian Optimization with Determinantal Point Processes. In Proceedings of The 25th International Conference on Artificial Intelligence and Statistics; PMLR, 2022; pp 7031–7054.
>
> [3] Wilson, J. T.; Moriconi, R.; Hutter, F.; Deisenroth, M. P. The Reparameterization Trick for Acquisition Functions. arXiv. 2017. arXiv:1712.00424.
>
> [4] Genz, A. Numerical Computation of Multivariate Normal Probabilities. Journal of Computational and Graphical Statistics 1992, 1 (2), 141–149.

---

> > ### Comment · Reviewer_tsqA · 2024-11-25
> >
> > Hi Authors,
> >
> > Thanks for the response.
> >
> > > If this statement were correct – that the biggest improvement comes from filtering 10,000 candidates – we would expect the random_10k baseline to perform similarly to pTS and qPO. However, we observe that the random_10k baseline performs poorly compared to other methods, indicating that filtering is not the source of the greatest improvement. We have based our implementations on BoTorch, where the memory footprint is limiting at these large candidate sizes, but agree that other approaches like variational GPs provide a path to address the larger candidate space, outside the scope of this work. We also note that this type of filtering due to computational cost is not unprecedented; Moss et al [1, Section 6.4] apply a similar pre-filtering to 1,000 compounds for their model-guided chemical library screen demonstration of GIBBON.
> >
> > I acknowledge that the authors are correct on this matter. My statement that the biggest improvement actually comes from greedy metric filtering was wrong. Previously, I mis-interpreted the meaning of the random selection baseline.
> >
> > > In our updated codebase, we include an implementation of our acquisition strategy by estimating orthant probabilities using Genz [4]. For our demonstrations in Section 4.2, we have tried implementing this and observe this method to be significantly more computationally expensive.
> >
> > I think there might be ways cache things smartly to bring down the complexity to \\(O(N^4)\\). Though, I have not spent time thinking thoroughly about this. So I will give authors that.
> >
> > Overall, I think the paper contains some simple (in a good way) and yet interesting ideas. But it feels like the paper needs a bit more development. So I think it's the best for the authors to revise and resubmit.

---

### Official Review · Reviewer_PfuF · 2024-11-07

**Soundness:** 2
**Presentation:** 3
**Contribution:** 2
**Rating:** 5
**Confidence:** 4

**Summary:**

The paper proposes an exploitive batched acquisition function that can be efficiently approximated as the sum of individual acquisitions. The method assumes a low-noise environment and explicitly encourages diversity in the queries. Due to the absence of an analytic form of the optimum, the acquisition function is implemented using Monte Carlo approximation. The paper demonstrates empirical evidence on real-world applications.

**Strengths:**

1. The core idea of the algorithm—devising a batch acquisition function that is the sum of individual acquisitions to enhance computational tractability—is convincing.

2. The illustrative case studies, along with figures, effectively demonstrate the advantages of promoting diversity.

**Weaknesses:**

1. The proposed algorithm resembles Predictive Thompson Sampling (pTS). Although the paper briefly discusses the differences between the proposed qPO and pTS and argues its advantages over existing methods, I find the argument not sufficiently convincing. There are two main reasons: First, pTS comes with convergence guarantees, whereas qPO only provides experimental results on unconventional metrics. Second, the randomness inherent in pTS and qEI is arguably the merit, offering robustness to model misspecification—which is common in real-world applications, as the paper also mentions. I hope the authors can elaborate more on the source of robustness in qPO, whether it arises from the sampling-based approximation or the deterministic acquisition that encourages diversity.

2. The paper omits a critical baseline: GIBBON (Moss et al., 2021), which shares the advantage of efficient batched acquisition calculation and is easy to implement with BoTorch.

3. Despite claiming that qPO is designed to be a computationally more efficient batched acquisition function, the paper includes no comparison of computational complexity, either analytically or empirically. This omission is problematic, especially since a Monte Carlo approximation is employed.

4. The metrics used in Section 4 are not the conventional cumulative regret or simple regret. Instead, the paper reports the retrieval of the top-k and the average reward of the top-k. It's unclear if these are valid choices without further justification.


***References***

- https://botorch.org/tutorials/GIBBON_for_efficient_batch_entropy_search

- Moss, Henry B., David S. Leslie, Javier Gonzalez, and Paul Rayson. "Gibbon: General-purpose information-based bayesian optimisation." Journal of Machine Learning Research 22, no. 235 (2021): 1-49.

**Questions:**

1. I'm a bit confused by the discussion regarding the ensemble method as a surrogate model. It seems that the experimental section employed a Gaussian Process with a Tanimoto kernel. What is the role of the ensemble method in the paper, especially as shown in Figure 1?

2. Why is the equality between Equation (6) and Equation (7) valid? What are the assumptions about $f$ and $y$? It is confusing that the paper assumes $f$ is noise-free in line 87 while also assuming low but non-zero observation noise for the GP in line 168. Is there the assumption that $f$ follows a certain distribution, e.g., GP? If not, what is the distribution of $f$?

---

> ### Author Response · Authors · 2024-11-19
> **Authors' Response to Reviewer PfuF (part 1 of 3)**
>
> We thank the reviewer for their thoughtful feedback on our submission. We have included our responses to the reviewer's comments and questions below. If there are any additional comparisons or clarifications in the text that might improve your impression and assessment of our work, please do not hesitate to let us know.
>
> **Weaknesses**
>
> > 1. The proposed algorithm resembles Predictive Thompson Sampling (pTS). Although the paper briefly discusses the differences between the proposed qPO and pTS and argues its advantages over existing methods, I find the argument not sufficiently convincing. There are two main reasons: First, pTS comes with convergence guarantees, whereas qPO only provides experimental results on unconventional metrics.
>
> We agree that deriving convergence guarantees could support the use of qPO, however we think it is a mistake to view the absence of convergence guarantees as a flaw of qPO, because:
> 1. Convergence guarantees for other algorithms usually depend on idealized assumptions which may not hold in practice. For example, bounds derived for TS-RSR [3] and pTS [11] utilize bounds derived by Srinivas et al [12], which assume $f$ either is sampled from a known Gaussian process or has a bounded norm in a known RKHS.
> 2. Convergence guarantees are almost always upper bounds, which may not be tight
> 3. These bounds are usually quite loose for small numbers of iterations, which is the scenario targeted by qPO
>
> We argue that in our setting, qPO should mainly be judged by its empirical results.
>
> We have included empirical results for cumulative regret, a conventional metric for the assessment of acquisition functions. We discuss this addition in more detail in the response to Weakness 4.
>
> > Second, the randomness inherent in pTS and qEI is arguably the merit, offering robustness to model misspecification—which is common in real-world applications, as the paper also mentions. I hope the authors can elaborate more on the source of robustness in qPO, whether it arises from the sampling-based approximation or the deterministic acquisition that encourages diversity.
>
> Thank you for providing these thought-provoking comments. We would like to point out that qEI is not inherently random; some algorithms with closed-form solutions of qEI exist [1]. Like qPO, if the qEI acquisition function were optimized exactly, selections would be deterministic. Similarly, qPO’s robustness does not come from the sampling-based approximation we describe and implement. We believe qPO’s robustness stems from its ability to exploit points most likely to be optimal while also inherently promoting diverse selections, as exemplified by the example in Section 3.2. As we describe in Section 3.2 and supported by [2,3], parallel Thomspon sampling may fail to select diverse batches. We have modified Section 3.2 to convey this sentiment more clearly.
>
> > 2. The paper omits a critical baseline: GIBBON (Moss et al., 2021), which shares the advantage of efficient batched acquisition calculation and is easy to implement with BoTorch.
>
> We have updated our first evaluation to include the GIBBON baseline. However, GIBBON led to a memory error in our second empirical study. We are currently working to resolve this, but it is also worth noting that practical computational complexity is an important perspective when comparing the merits of BO methods.

---

> ### Author Response · Authors · 2024-11-19
> **Authors' Response to Reviewer PfuF (part 2 of 3)**
>
> **Weaknesses (continued)**
>
> > 3. Despite claiming that qPO is designed to be a computationally more efficient batched acquisition function, the paper includes no comparison of computational complexity, either analytically or empirically. This omission is problematic, especially since a Monte Carlo approximation is employed.
>
> We do not claim that qPO is more computationally efficient than other batch acquisition functions, only that it does not require sequential (myopic) construction. This sequential batch construction policy used by qEI and qPI, for example, may result in the selection of suboptimal batches. qPO does not suffer this limitation. We would expect this distinction to result in improved performance in terms of sample efficiency, not necessarily computational efficiency. We now include a comparison of wall time for running qPO and baselines in Appendix Section A.5 (also pasted below).
>
> | Method    | Wall time for QM9 experiment (s) | Wall time for antibiotics experiment (s) |
> |-----------|----------------------------------|------------------------------------------|
> | qPO       | 3774 $\pm$ 19                    | 1883 $\pm$ 9                             |
> | pTS       | 3715 $\pm$ 19                    | 1855 $\pm$ 13                            |
> | qEI       | 6016 $\pm$ 9                     | 1974 $\pm$ 3                             |
> | Greedy    | 46 $\pm$ 1                       | 20 $\pm$ 0                               |
> | UCB       | 51 $\pm$ 8                       | 26 $\pm$ 8                               |
> | TS-RSR    | 3687 $\pm$ 18                    | 1834 $\pm$ 13                            |
> | BUCB      | 4840 $\pm$ 12                    | 1401 $\pm$ 6                             |
> | qPI       | 4850 $\pm$ 12                    | 1385 $\pm$ 6                             |
> | DPP-TS    | 5728 $\pm$ 43                    | 2902 $\pm$ 36                            |
> | random10k | 44 $\pm$ 2                       | 20 $\pm$ 2                               |
>
>
> > 4. The metrics used in Section 4 are not the conventional cumulative regret or simple regret. Instead, the paper reports the retrieval of the top-k and the average reward of the top-k. It's unclear if these are valid choices without further justification.
>
> Top-k metrics are common for assessing model-guided chemical library searches [4-8], as the primary goal of these accelerated searches is to efficiently identify the top-performing compounds. In contrast to other BO applications, chemical library exploration typically aims to identify a set of top-performing designs to test, instead of solely identifying a single global optimum. Therefore, assessment of not only the very best acquired point (as in regret) is preferred for molecular design applications. Because it is possible (but inefficient) to screen the entire library exhaustively, the top-k metric provides an accurate indication of the efficiency gains of using model-guided BO. This is in contrast to continuous BO, where it is impossible to exhaustively screen every candidate design. Additionally, because of the discrete nature of the design space, the top-1 compound may have a significantly better score than the second best compound. Therefore, regret metrics these applications are more noisy metrics and less informative of improved efficiency. Nonetheless, we now include comparisons according to cumulative regret in Section 4.
>
> **Questions**
>
> > 1. I'm a bit confused by the discussion regarding the ensemble method as a surrogate model. It seems that the experimental section employed a Gaussian Process with a Tanimoto kernel. What is the role of the ensemble method in the paper, especially as shown in Figure 1?
>
> As correctly stated by the reviewer, no ensemble is used in the experimental section. In Figure 1, we aim to highlight that qPO can be applied to model architectures that do not explicitly output prediction means and covariances. For example, if a model ensemble is trained, a multivariate Gaussian can be modeled on ensemble predictions, allowing for the use of qPO acquisition. While we do not include such an application in our experiments, we found it important to underscore that qPO is not limited to Gaussian process surrogates, as other surrogate architectures like ensembles or MC dropout are common for uncertainty quantification in molecular property prediction applications [4, 9, 10].

---

> ### Author Response · Authors · 2024-11-19
> **Authors' Response to Reviewer PfuF (part 3 of 3)**
>
> **Questions (continued)**
>
> > 2. Why is the equality between Equation (6) and Equation (7) valid? What are the assumptions about $f$ and $y$? It is confusing that the paper assumes $f$ is noise-free in line 87 while also assuming low but non-zero observation noise for the GP in line 168. Is there the assumption that $f$ follows a certain distribution, e.g., GP? If not, what is the distribution of $f$?
>
> We have modified these equations to better represent $f$ and $y$, which were not precisely defined in our original submission. We now consider $f$ to be the ground-truth objective value, while $y$ represents predictions from the model posterior. We ask the reviewer to please review the changes to Section 2.2 in the updated submission.
>
> **Citations**
>
> [1] Chevalier, C.; Ginsbourger, D. Fast Computation of the Multi-Points Expected Improvement with Applications in Batch Selection. In Learning and Intelligent Optimization; Nicosia, G., Pardalos, P., Eds.; Springer: Berlin, Heidelberg, 2013; pp 59–69.
>
> [2] Nava, E.; Mutny, M.; Krause, A. Diversified Sampling for Batched Bayesian Optimization with Determinantal Point Processes. In Proceedings of The 25th International Conference on Artificial Intelligence and Statistics; PMLR, 2022; pp 7031–7054.
>
> [3] Ren, Z.; Li, N. TS-RSR: A Provably Efficient Approach for Batch Bayesian Optimization. arXiv 2024. arXiv:2403.04764
>
> [4] Yang, Y.; Yao, K.; Repasky, M. P.; Leswing, K.; Abel, R.; Shoichet, B. K.; Jerome, S. V. Efficient Exploration of Chemical Space with Docking and Deep Learning. J. Chem. Theory Comput. 2021, 17 (11), 7106–7119.
>
> [5] Gao, W.; Fu, T.; Sun, J.; Coley, C. W. Sample Efficiency Matters: A Benchmark for Practical Molecular Optimization. In Thirty-sixth Conference on Neural Information Processing Systems Datasets and Benchmarks Track; 2022.
>
> [6] Pyzer-Knapp, E. O. Bayesian Optimization for Accelerated Drug Discovery. IBM Journal of Research and Development 2018, 62 (6), 2:1-2:7.
>
> [7] Wang-Henderson, M.; Soyuer, B.; Kassraie, P.; Krause, A.; Bogunovic, I. Graph Neural Bayesian Optimization for Virtual Screening. In NeurIPS 2023 Workshop on Adaptive Experimental Design and Active Learning in the Real World; 2023.
>
> [8] Cao, Z.; Sciabola, S.; Wang, Y. Large-Scale Pretraining Improves Sample Efficiency of Active Learning Based Molecule Virtual Screening; In NeurIPS 2023 Workshop on New Frontiers of AI for Drug Discovery and Development; 2023.
>
> [9] Jiang, S.; Qin, S.; Lehn, R. C. V.; Balaprakash, P.; M. Zavala, V. Uncertainty Quantification for Molecular Property Predictions with Graph Neural Architecture Search. Digital Discovery 2024, 3 (8), 1534–1553.
>
> [10] Wen, M.; Tadmor, E. B. Uncertainty Quantification in Molecular Simulations with Dropout Neural Network Potentials. npj Comput Mater 2020, 6 (1), 1–10.
>
> [11] Kandasamy, K.; Krishnamurthy, A.; Schneider, J.; Poczos, B. Parallelised Bayesian Optimisation via Thompson Sampling. In Proceedings of the Twenty-First International Conference on Artificial Intelligence and Statistics; PMLR, 2018; pp 133–142.
>
> [12] Srinivas, N.; Krause, A.; Kakade, S.; Seeger, M. Gaussian Process Optimization in the Bandit Setting: No Regret and Experimental Design. In Proceedings of the 27th International Conference on International Conference on Machine Learning; ICML’10; Omnipress: Madison, WI, USA, 2010; pp 1015–1022.

---

> > ### Comment · Reviewer_PfuF · 2024-11-25
> >
> > I appreciate the authors' detailed rebuttal and acknowledge that the revisions have improved the clarity and comprehensiveness of the experimental section. However, some concerns remain unaddressed:
> >
> > 1.  The authors argue that the proposed algorithm alleviates the combinatorial optimization challenges of non-myopic algorithms and employs filtering to improve the traceability of acquisition optimization. However, the claimed benefits are not sufficiently reflected in either optimization performance or efficiency when compared to existing methods, particularly after incorporating the latest batch of results.
> >
> > 2. I share reviewer ***gD5x’s*** concern regarding soundness in the current version. Specifically, while similar existing methods provide regret-based justification, this paper does not clearly differentiate itself either in terms of the assumptions it relies on or through a detailed discussion of the performance metrics where qPO demonstrates better results. Although the authors attempt to differentiate assumptions between $f$ and $\hat{f}$, the acting surrogate model $\hat{f}$ still appears to inherit similar assumptions as previous methods that rely on Gaussian processes.
> >
> > 3. The authors suggest that the robustness of qPO arises from its ability to exploit points most likely to be optimal rather than relying on randomness. However, the manuscript indicates that a modest number of $M$ is used in the sampling process and does not provide sufficient evidence to separate the contributions of the sampling procedure from those of the acquisition function's design. I encourage the authors to include additional evidence or insights to substantiate this claim.

---

### Author Response · Authors · 2024-11-19
**Summary of Responses to All Reviewers**

We appreciate the valuable comments and suggestions provided by the reviewers. We have addressed each point and revised our manuscript accordingly. Below, we summarize the key changes and address common questions raised by the reviewers.

**Highlights of Changes**

1. **Expanded Baseline Comparisons**: We incorporated additional baselines into our evaluation, including GIBBON [1], TS-RSR [2], DPP-TS [3], batch UCB [4], and qPI, as suggested by reviewers. These updates provide a more comprehensive empirical comparison, which demonstrates the competitive performance of qPO. Furthermore, we provided empirical results for cumulative regret and wall time comparisons.

2. **Improved Clarity of Mathematical Exposition**: We clarified the mathematical exposition, including the definitions of $\textbf{x}$, $f$, and $y$, and provided additional insights into how qPO differs from existing methods such as qPI and qEI. We revised Section 2.2 and Algorithm 1 for clarity and updated the paper title to reflect the primary contribution more accurately.

**Key Insights and Contributions**

1. **Theoretical Convergence Analysis Omission**: Several reviewers highlighted the lack of theoretical convergence guarantees. While we acknowledge this as a limitation, we argue that it is adequate to judge qPO based on empirical results, because:
     - Convergence guarantees for other algorithms usually depend on idealized assumptions which may not hold in practice. For example, bounds derived for TS-RSR [2] and pTS [5] utilize bounds derived by Srinivas et al [6], which assume $f$ either is sampled from a known Gaussian process or has a bounded norm in a known RKHS.
    - Convergence guarantees are almost always upper bounds, which may not be tight
    - These bounds are usually quite loose for small numbers of iterations, which is the scenario targeted by qPO

2. **Novel Acquisition Function**: The qPO method introduces a principled approach to batched Bayesian optimization by maximizing the probability of including the optimum. Despite being motivated by exploitation, this acquisition strategy exhibits explorative behavior by promoting diversity, without requiring myopic batch construction.

3. **Application-Specific Relevance**: The use of qPO in finite discrete domains, such as chemical library exploration, addresses practical challenges in molecular design. By optimizing for top-k performance rather than regret metrics, qPO aligns with the objectives of real-world applications.

We thank the reviewers for their time and insightful input, which have been invaluable in refining our work. We hope that these changes have addressed the concerns and improved your assessment of our contributions.

**Citations**

[1] Moss, H. B.; Leslie, D. S.; González, J.; Rayson, P. GIBBON: General-Purpose Information-Based Bayesian Optimisation. J. Mach. Learn. Res. 2021, 22 (1), 235:10616-235:10664.

[2] Ren, Z.; Li, N. TS-RSR: A Provably Efficient Approach for Batch Bayesian Optimization. arXiv 2024. arXiv:2403.04764.

[3] Nava, E.; Mutny, M.; Krause, A. Diversified Sampling for Batched Bayesian Optimization with Determinantal Point Processes. In Proceedings of The 25th International Conference on Artificial Intelligence and Statistics; PMLR, 2022; pp 7031–7054.

[4] Wilson, J. T.; Moriconi, R.; Hutter, F.; Deisenroth, M. P. The Reparameterization Trick for Acquisition Functions. arXiv. 2017. arXiv:1712.00424.

[5] Kandasamy, K.; Krishnamurthy, A.; Schneider, J.; Poczos, B. Parallelised Bayesian Optimisation via Thompson Sampling. In Proceedings of the Twenty-First International Conference on Artificial Intelligence and Statistics; PMLR, 2018; pp 133–142.

[6] Srinivas, N.; Krause, A.; Kakade, S.; Seeger, M. Gaussian Process Optimization in the Bandit Setting: No Regret and Experimental Design. In Proceedings of the 27th International Conference on International Conference on Machine Learning; ICML’10; Omnipress: Madison, WI, USA, 2010; pp 1015–1022.

---

### Meta-Review · Area_Chair_NPcB · 2024-12-29

**Metareview:**

Motivated by molecular design problems, where discrete designs (from some fixed database) can be evaluated in parallel, this work proposes an acquisition function for batch BO in discrete settings. Reviewers appreciate the simplicity of the approach, no reviewer advocated for acceptance, but provide encouraging remarks to revise and resubmit to another venue.

Some main areas of improvement include:

- Additional theoretical justification for the proposed AF. The authors pushed back in that such analysis would require assuming some particular known GP model classes, convergence would only be hold true asymptotically—whereas in BO we care about performance after a limited number of samples, etc. While I am personally not a fan of such proofs, such analysis is often expected by the community and can lead to insights.
- Clarity of exposition, motivation, and soundness of the proposed AF (see discussion with gD5x and concluding comments by PfuF)
- More conventional analysis. The authors consider only two test problems and empirical results on "unconventional metrics" (PfuF). While the authors note these are common to the molecule design community, the absence of simple regret wrt the best value found is missing.
- Other real or synthetic benchmarks from the literature could also be included (8R9v; the authors argue that these problems are too lowD, but there are other common discrete optimization benchmarks out there). -More rigorous analysis of performance characteristics, such as how simple regret or top-k mean scales with batch size, and sensitivity to the number of MC samples (8R9v)

Reviewers provide numerous constructive critiques of the work, ideas on how to improve the algorithm, and relevant references. With these improvements, this work can be quite compelling.

A few notes from my own reading of the paper:

- The authors frequently use the term "myopic" or "non-additive" acquisition functions. [1] (the expanded version of the workshop paper by Wilson et. al cited throughout the work) shows how many AFs can be solved via a sequential greedy approach due to submodularity. It feels odd to refer to such greedy algorithms as myopic as they can be shown to approximate the full joint optimization problem. Adopting some more of that language could be helpful.
- Table 1 should report ± 1.96sem (the 95% interval). It would be helpful to highlight cases where methods provide the best result by putting them in bold (which may result in multiple methods being bold, if the difference is not stat sig, which is the case for a number of the methods for various metrics).
- Why is qEI missing from fig3? It appeared to be a top contender in Table 1.
- The authors should clearly define what "hallucination" means (as it is less commonly used)—essentially just sequentially conditioning on the mean value. It should be distinguished it from the use of "fantasies" in sequential greedy optimization MC variants of AFs, which is core to many of the results in [1]. This approach is utilized heavily in BoTorch's parallel AFs, including qPI and the various qEI variants. Such fantasization can also be used to implement smarter versions of TS (perhaps that is what pTS is?).
- In general it is unclear on whether the contribution of specific to discrete spaces, or would work for continuous or mixed spaces. -If the contribution is specific to discrete spaces, other discrete problems and choices of kernels could be helpful.
- [2] proposes an approach to solving similar problems that induces diversity in the input space. I have no idea how well it works in the smaller sample regime, but perhaps some of the baseline tasks or ways of reporting the results could be useful (and direct comparison could make the results more compelling)

[1] Maximizing acquisition functions for Bayesian optimization. Wilson, Hutter, Deisenroth, NeurIPS 2018.

[2] Discovering Many Diverse Solutions with Bayesian Optimization. Maus et al. AISTATS 2023.

**Additional Comments On Reviewer Discussion:**

Many reviewers provided detailed feedback to the reviewers, which resulted in some clarification of exposition and additional experiments. These arguments and results did not move the needle on acceptance. The reviews were constructive, and pointed to several promising directions and ways to have this work better tie into the literature.

---

### Decision · Program_Chairs · 2025-01-22

Reject